# Mitochondria isolated from lipid droplets of white adipose tissue reveal functional differences based on lipid droplet size

Alexandra J Brownstein[1,2] , Michaela Veliova[1] , Rebeca Acin-Perez[1] , Frankie Villalobos[1], Anton Petcherski[1] , Alberto Tombolato[1], Marc Liesa[1,3,*] , Orian S Shirihai[1,2,*]

**Recent studies in brown adipose tissue (BAT) described a unique subpopulation of mitochondria bound to lipid droplets (LDs), which were termed PeriDroplet Mitochondria (PDM). PDM can be isolated from BAT by differential centrifugation and salt washes. Contrary to BAT, this approach has so far not led to the successful isolation of PDM from white adipose tissue (WAT). Here, we developed a method to isolate PDM from WAT with high yield and purity by an optimized proteolytic treatment that preserves the respiratory function of mitochondria. Using this approach, we show that, contrary to BAT, WAT PDM have lower respiratory and ATP synthesis capacities compared with WAT cytoplasmic mitochondria (CM). Furthermore, by isolating PDM from LDs of different sizes, we found a negative correlation between LD size and the respiratory capacity of their PDM in WAT. Thus, our new isolation method reveals tissue-specific characteristics of PDM and establishes the existence of heterogeneity in PDM function determined by LD size.**

## Introduction

Mitochondria attached to lipid droplets (LDs), or peridroplet mitochondria (PDM), were shown to expand lipid droplets in brown adipose tissue (BAT) (Benador et al, 2018). Mechanistically, BAT PDM are specialized to oxidize pyruvate and provide ATP to fuel the esterification of fatty acids into triglycerides (Benador et al, 2018). Whether this specialized function of PDM is conserved in tissues different from BAT and liver is unknown, mostly because of a lack of available protocols to isolate PDM efficiently from other tissues. The role of PDM promoting the esterification of fatty acids into triglycerides was hypothesized to be especially relevant in white adipose tissue (WAT), as esterification can protect from lipotoxicity of NEFA (non-esterified fatty acids) (Listenberger et al, 2003; Kuramoto et al,

2012; Wang et al, 2015; Laurens et al, 2016; Zheng et al, 2017; Tan et al, 2019; Veliova et al, 2020). It has been hypothesized that in WAT both PDM and cytoplasmic mitochondria (CM) contribute to the removal of NEFA, with CM oxidizing NEFA and PDM securing them into neutral triglycerides (Veliova et al, 2020). Finding ways in which we can manipulate or offset different mitochondrial populations has the potential to regulate lipid metabolism.

WAT has been shown to have lower mitochondrial mass per cell when compared with BAT. Moreover, per mitochondrial mass, WAT has a lower capacity to oxidize fatty acids when compared with BAT (Cannon & Nedergaard, 2004; Cinti, 2018). Nonetheless, recent studies have shown that mitochondria from WAT not only support adipocyte-specific functions, but play essential roles in maintaining whole-body energy homeostasis, control of insulin sensitivity, glucose metabolism, and crosstalk between muscles and adipose tissues (Boudina & Graham, 2014; Vernochet et al, 2014; Lee et al, 2019). Electron microscopy images of human WAT support the existence of a unique population of mitochondria that are in contact with lipid droplets in WAT; however, the function of these mitochondria within the cell, and the changes associated with the development of obesity are still unknown (Cushman, 1970; Cinti, 2018; Freyre et al, 2019). To better understand the role of PDM in WAT, it is crucial to develop an optimized protocol that allows for PDM isolation with a high yield.

The PDM isolation protocols published to date use differential centrifugation. However, these protocols differ in the composition of the homogenization and isolation buffers used, more specifically in salt concentrations and the presence of detergents (Yu et al, 2015; Zhang et al, 2016; Cui et al, 2019; Ngo et al, 2021). Benador et al showed successful stripping of a large proportion of mitochondria attached to lipid droplets in BAT by centrifuging the fraction enriched with lipid droplets at high speed (Benador et al, 2018; Ngo et al, 2021). However, not all lipid-bound mitochondria were removed by this centrifugation (Benador et al, 2018). This may reflect on the heterogeneity of the mechanisms adhering mitochondria to

[1]David Geffen School of Medicine, Department of Medicine (Endocrinology) and Department of Molecular and Medical Pharmacology, University of California, Los Angeles, CA, USA  [2]Molecular Cellular Integrative Physiology Interdepartmental Graduate Program, University of California, Los Angeles, CA, USA  [3]Department of Cells and Tissues, Institut de Biologia Molecular de Barcelona, IMBB, CSIC, Barcelona, Spain

Correspondence: mlrbmc@ibmb.csic.es; oshirihai@mednet.ucla.edu
*Marc Liesa and Orian S Shirihai contributed equally to this work

lipid droplets and may represent functional diversity of PDM. To explore the diversity of PDM, a more powerful approach to detach mitochondria from lipid droplets needs to be applied. We rationalized that digestion of the proteins that link mitochondria to LDs could enhance the detachment of PDM from lipid droplets, an approach that would require addressing the potential damage induced by the proteolytic activity. Accordingly, several studies in oxidative tissues, including skeletal and cardiac muscle, used protease treatments to disperse the tissue in the process of mitochondrial isolation without impairing their oxidative function (Kras et al, 2016; Lai et al, 2019; Sánchez-González & Formentini, 2021).

Here, we describe a new approach to isolate PDM that combines protease treatment and centrifugation. The combined protease and centrifugation method successfully detaches mitochondria from WAT lipid droplets, revealing that BAT and WAT PDM differ in the strength of attachment to lipid droplets and in their bioenergetic characteristics. Moreover, using our new approach, we detached PDM from small and large lipid droplets, uncovering the functional diversity of PDM segregated by their lipid droplet size.

# Results

### The attachment of mitochondria to lipid droplets is stronger in WAT compared with BAT

Previously published protocols to isolate PDM from BAT demonstrated that the centrifugal force applied to lipid droplets (LDs) was sufficient to strip a significant portion of the PDM in BAT (Fig 1C and D) (Benador et al, 2018; Acín-Perez et al, 2021; Ngo et al, 2021). In WAT, PDM isolation was historically low. By using our previously published protocol to image mitochondria in fat layers (Acín-Perez et al, 2021), we found that there is a major fraction of PDM in WAT fat layers that remain attached to the LDs after centrifugation (Fig 1A and B). This result suggested that the attachment of PDM to LDs in WAT is more resistant to centrifugation than in BAT, highlighting that the interaction between PDM and LDs is stronger in WAT.

### A WAT-specific protein–protein or protein–lipid interaction may explain the stronger attachment of PDM in WAT versus BAT

Our results show that mechanical separation effectively removed the majority of, but not all, PDM from lipid droplets in the BAT (Fig 1C and D). However, application of the same mechanical protocol to WAT did not result in any significant removal of PDM from lipid droplets (Fig 1A and B), suggesting the possibility that the composition and/or abundancy of tethers between mitochondria and lipid droplets are different in BAT and WAT. We hypothesized that the resistance to removal by centrifugation may come from either protein–protein or protein–lipid interactions. Independent of which of the above mechanisms contribute to mitochondria–LD tethering, a protein is expected to be involved and therefore should be sensitive to proteolytic activity. To test our hypothesis, we treated the fat layers of WAT with Proteinase K (Prot K), to digest the protein-mediated tethers anchoring mitochondria to LDs and

potentially strip mitochondria that are resistant to stripping by centrifugation.

To confine degradation to the protein tethers in the LD and outer mitochondrial membrane, we inactivated Prot K with PMSF right before separating PDM from the LDs by centrifugation (Fig 1E) (Gold & Fahrney, 1964; Gold, 1965; Badugu et al, 2008; Koncsos et al, 2018). Furthermore, PMSF addition would allow long-term storage of the fractions by freezing them, as Prot K is still active after freeze and thaw cycles. Prot K is widely used to identify the domains of outer mitochondrial membranes exposed to the cytosol, as Prot K cannot diffuse across intact mitochondrial membranes. Therefore, Prot K cannot degrade matrix, inner membrane proteins or even integral outer membrane proteins facing the intermembrane space in intact mitochondria; only if membranes are pierced by freeze–thawing (Badugu et al, 2008; Denuc et al, 2016).

To establish the efficacy of protease treatment on the yield and function of isolated PDM, we performed paired comparisons of protease-treated LD fractions with their respective controls. To this end, we split the WAT homogenates into four groups (Fig 1E): in the first group, PDM were isolated exclusively by centrifuging the LD fraction as was previously published (Benador et al, 2018). In the second group, the LD fraction was treated with Prot K before centrifugation, and in the third group, the LD fraction was treated with Prot K and then incubated with PMSF before centrifugation. Lastly, the fourth fraction was treated only with PMSF, to establish whether PMSF itself could change PDM isolation and function. As an additional control, we added Prot K to the supernatant from which cytosolic mitochondria (CM) are isolated (Fig 1E). The rationale was that cytosolic mitochondria were expected to have fewer protein tethers and thus Prot K actions would be harder to confine on CM.

We found that Prot K treatment significantly increased the amount of protein in the PDM fraction of WAT, demonstrating an improvement in the yield of PDM isolation (Fig 1F). Prot K treatment also increased the yield of CM isolated from WAT (Fig 1F), suggesting that the CM in WAT were potentially tethered via protein-mediated interactions to other membranes (ER or nucleus) that were disrupted with Prot K.

Isolation of PDM from BAT using the centrifugation method, whereas being efficient, still left some mitochondria attached to the lipid droplet as shown above in Fig 1C and D. Intrigued by the results obtained with Prot K in the WAT, we decided to test Prot K in BAT. In BAT, we also found that Prot K treatment significantly increased the yield of PDM isolation, but not of BAT CM (Fig 1G). These data suggest that both WAT PDM and WAT CM potentially have different mechanisms regulating their tethering to other organelles and membranes.

We next sought to confirm whether the increase in total protein observed in the PDM fraction was associated with an increase in mitochondrial content, and with an increase in the purity of mitochondrial fractions. To determine purity, we used Western blot to quantify the amount of mitochondrial proteins per microgram of protein in PDM and CM fractions isolated from WAT and BAT. As PDM isolation is a long procedure, we could not perform the Western blots in freshly isolated mitochondria. Complicating the analysis of frozen samples, proteins inserted in the outer membrane (VDAC), and inner membrane proteins (OXPHOS) can be degraded by Prot K when mitochondrial membranes are damaged by freeze–thaw cycles (Badugu et al, 2008; Zhang et al, 2015).

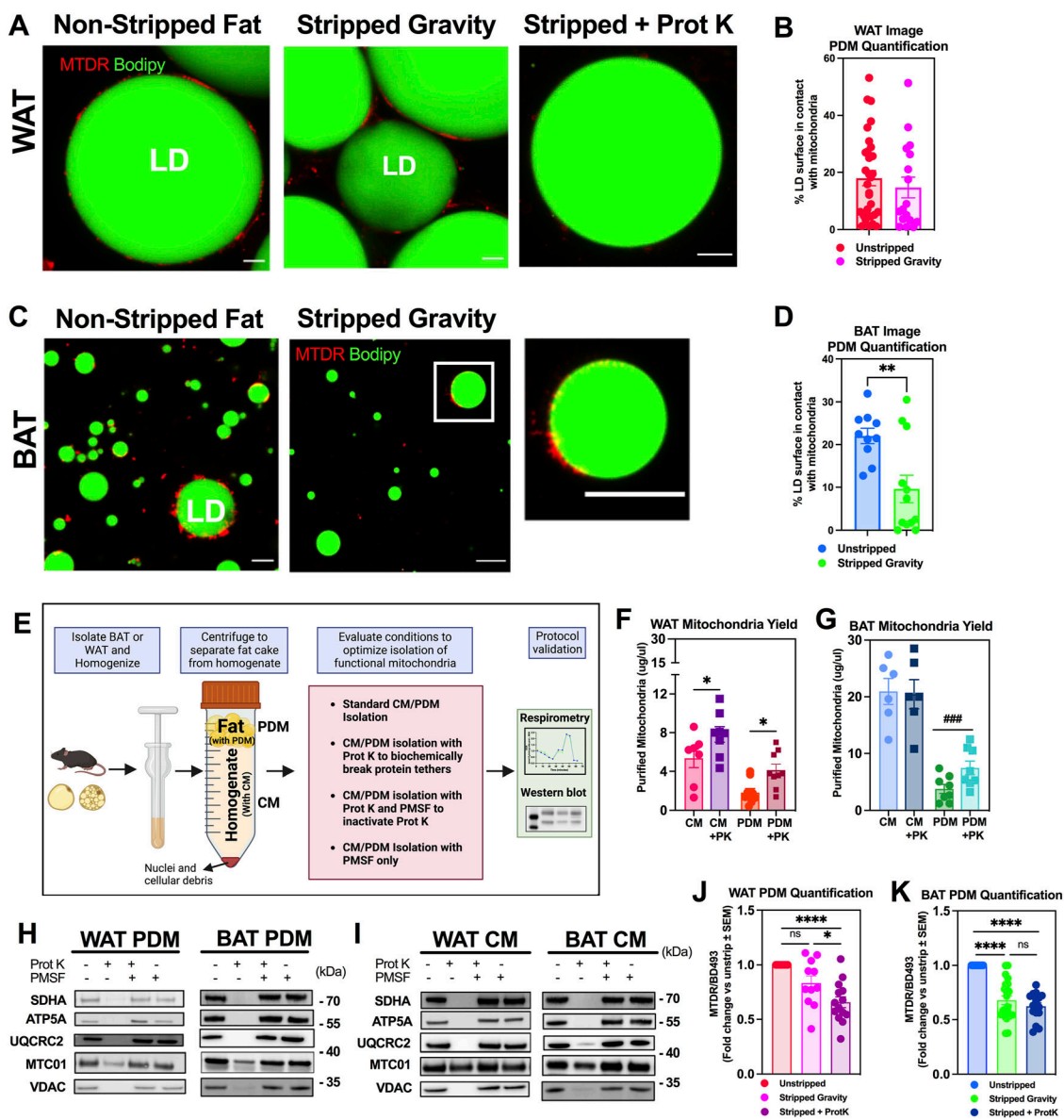

**Figure 1. Proteinase K treatment enables the isolation of peridroplet mitochondria (PDM) from white adipose tissue (WAT) and increases the yield of PDM isolated from brown adipose tissue (BAT).**

**(A)** Super-resolution confocal microscopy of lipid droplet fractions from WAT, visualizing PDM, (red) attached to lipid droplets (LDs; green) before (Non-stripped Fat) after high-speed centrifugation (Stripped Fat), and after proteinase K (Prot K) treatment and high-speed centrifugation (Stripped + Prot K). Mitochondria were stained with Mitotracker DeepRed (MTDR) and LDs with BODIPY493/503 (BD493). Scale Bar = 10 microns. **(B)** PDM quantification of WAT LDs determined by quantifying the % of LD surface covered by mitochondria from images shown in panel (A). 18–32 lipid droplets were analyzed from four independent isolations. **(C)** Imaging and staining performed as in (A) but using LD fractions from BAT. **(D)** PDM quantification of images from BAT shown in panel (C). Dot plot shows quantification of the % of LD surface covered by mitochondria as in panel (B). 10–12 images were analyzed, each with multiple lipid droplets from five independent isolations. For each image, the total % of LD surface covered by mitochondria was averaged and represents a single point on the graph. **(E)** Summary of new method enabled by Prot K to isolate PDM and cytoplasmic (CM) mitochondria from WAT and BAT. Fat homogenates were centrifuged at low speed to separate the fat layer containing PDM from supernatant containing the CM. Prot K was added to both the LD fraction and supernatant and incubated for 15 min. High-speed centrifugation separated PDM from Prot K–treated lipid droplets and CM from the supernatant. **(F, G)** Comparison of protein recovered in PDM and CM fractions in the presence or absence of Prot K treatment from (F) WAT and (G) BAT. N = 6–9 independent isolation experiments, with each individual data point representing one single experiment. **(H, I)** Western blot of PDM (H) and CM (I) isolated from WAT and BAT with regular PDM protocol, PDM isolation + Prot K, PDM isolation with Prot K + PMSF and PMSF alone. N = 3–4 independent experiments of PDM and CM isolation. **(J, K)** PDM quantification of WAT (J) and BAT (K) using the fluorescence plate reader assay. Mitochondria were stained with MTDR and LDs with BD493. The ratio of MTDR/BD493 fluorescence was quantified as a measure of PDM mass per LD mass. Five to six independent isolation experiments. *$P < 0.05$, **$P < 0.01$, ***$P < 0.001$, ****$P < 0.0001$ by one-way ANOVA. ###$P < 0.001$ compared with untreated CM or PDM by $t$ test. All data presented as mean ± SEM. Illustrations were created in BioRender.

In agreement with these published findings, we found that the content of VDAC and OXPHOS subunits was dramatically decreased in PDM and CM fractions frozen and thawed after isolation from WAT and BAT that did not contain PMSF (Fig 1H and I). Treatment with PMSF prevented the decreases in VDAC and OXPHOS protein content and preserved the proteins of freeze–thawed mitochondria isolated with Prot K (Fig 1H and I). Quantification of VDAC and OXPHOS proteins by Western blot revealed no differences in their content per microgram of protein (Fig S1A–D), which showed that Prot K treatment proportionally increased the yield of mitochondrial and contaminant proteins in PDM fractions. The effects of PMSF confirm that the decrease in VDAC and OXPHOS content was caused by the persistence of Prot K activity in PDM and CM fractions even after washes and freeze–thawing. Finally, PMSF treatment by itself did not result in any significant differences in VDAC and OXPHOS protein content (Fig S1A–D). This latter result supports that the inhibitory actions of PMSF on respiration were not caused by a decrease in OXPHOS protein content or the purity of the isolated fraction.

The finding that Prot K treatment facilitates the separation of mitochondria from LDs was further confirmed by our plate reader-based assay quantifying PDM (Benador et al, 2018; Acín-Perez et al, 2021; Ngo et al, 2021). The lipid droplet fractions were stained with BODIPY493/503 (BD493), a dye staining neutral lipids, and Mito-Tracker DeepRed (MTDR), a dye staining mitochondria (Acín-Perez et al, 2021). We chose to use MTDR because we have previously verified that its membrane potential dependency is minimal and does not impose a significant bias over mitochondrial mass with the concentration and duration of staining required (Acín-Perez et al, 2021). To evaluate the amount of mitochondria per LD, the LD fractions were stained before and after separating the mitochondria. The reduction in the ratio of MTDR/BD493 fluorescence after separating the mitochondria by centrifugation was used to quantify PDM content (Acín-Perez et al, 2021). In WAT, centrifugation alone was unable to strip a significant amount of PDM (Fig 1J), confirming the confocal imaging, where we see PDM retained after centrifugation (Fig 1A and B). We were only able to strip the PDM from WAT when LD fractions were treated with Prot K (Fig 1J). In BAT, the ratio of MTDR/BD493 was significantly reduced by centrifugation alone and was further reduced, although not significantly, when Prot K was added (Fig 1K). Together these data suggest that Prot K treatment enhances PDM isolation from WAT, and can increase the yield of PDM isolated from BAT.

## Isolation of PDM with Prot K does not inhibit mitochondrial respiratory function

Despite Prot K not being able to access the inner membrane of intact mitochondria, both PMSF and Prot K treatments were reported to impact oxygen consumption measured in intact mitochondria (Marcillat et al, 1988; Ruiz-Meana et al, 2014; Cole et al, 2015; Koncsos et al, 2018). Thus, we wanted to verify whether our treatments with Prot K, without PMSF, were affecting respiratory function of freshly isolated mitochondria. In addition, we determined whether PMSF treatment at 2 mM affected mitochondrial respiratory function by itself. Lower concentrations (below 2 mM) of PMSF were previously shown to have marginal or no effects on

mitochondrial respiration (Koncsos et al, 2018), but these lower concentrations of PMSF cannot block Prot K effectively.

Respiratory function of isolated PDM and CM using Prot K was determined by quantifying oxygen consumption rates (OCR) using the XF96 flux analyzer. We measured state 3 respiration, which reflects coupled respiration associated with maximal ATP synthesis, and separately, maximal complex IV (CoxIV) activity. CoxIV activity is measured by providing N,N,N′,N′-Tetramethyl-p-phenylenediamine (TMPD)/Ascorbate directly to isolated mitochondria, as TMPD donates electrons directly to complex CoxIV. Prot K ± PMSF-treated CM and PDM respiration data were normalized as fold change to respiration of untreated mitochondria isolated from the same tissue sample.

A significant increase in state 3 OCR was observed in WAT CM isolated using Prot K (Fig 2A). PMSF treatment prevented the increase in state 3 induced by Prot K treatment. However, PMSF alone decreased respiration rates to the same levels as mitochondria treated with Prot K + PMSF. Therefore, the decrease associated with PMSF is explained by an autonomous effect of PMSF decreasing state 3 respiration, rather than by blocking protein degradation. A similar inhibitory effect of PMSF on respiration was observed when providing electrons directly to CoxIV (Fig 2B). On the other hand, CoxIV-driven respiration was not increased in mitochondria isolated using Prot K, supporting the notion that increased state 3 respiration reflected a specific increase in ATP-synthesizing activity. To confirm this possibility, we measured ATP synthesis rates in isolated mitochondria using firefly luciferase luminescence. We find that Prot K treatment increased ATP synthesis rates in WAT CM (Fig 2C), further supporting the idea that the addition of Prot K results in a mitochondrial fraction enriched with functional, non-damaged mitochondria.

The effects of Prot K on WAT PDM respiratory function showed some similarities to WAT CM. Isolating PDM using Prot K increased both state 3 respiration (Fig 2D) and ATP synthesis rates (Fig 2F), as in WAT CM. PMSF resulted in decreased ATP synthesis rates in PDM as well. The major difference induced by Prot K treatment in PDM was an increase in CoxIV activity, which was not observed in CM (Fig 2E). In all, isolating either CM or PDM using Prot K yielded mitochondria with higher respiratory capacity.

Because we observed that Prot K treatment selectively increased ATP synthesizing respiration in some mitochondrial preparations, it was still possible that Prot K induced mild damage to the outer membrane, to cause mild cytochrome C leakage. If mild leakage was present, we should see a larger increase in respiration induced by cytochrome C supplementation in mitochondria isolated with Prot K, when compared with cytochrome c supplementation in mitochondria isolated without Prot K. It is important to note that cytochrome c supplementation can increase oxygen consumption independently of complex IV activity as well. We found that cytochrome c supplementation did not induce a larger increase in respiration in Prot K-treated mitochondria, with this effect being even of lower magnitude than the effect in control mitochondria (Fig S2A and C). In this regard, the largest portion of cytochrome c-induced increase in respiration observed in both control and Prot K-treated mitochondria was preserved in azide-treated mitochondria (Fig S2B and D). This latter result shows that cytochrome c-induced increase in respiration is not explained by an increase in the availability of cytochrome c for mitochondrial respiration. These

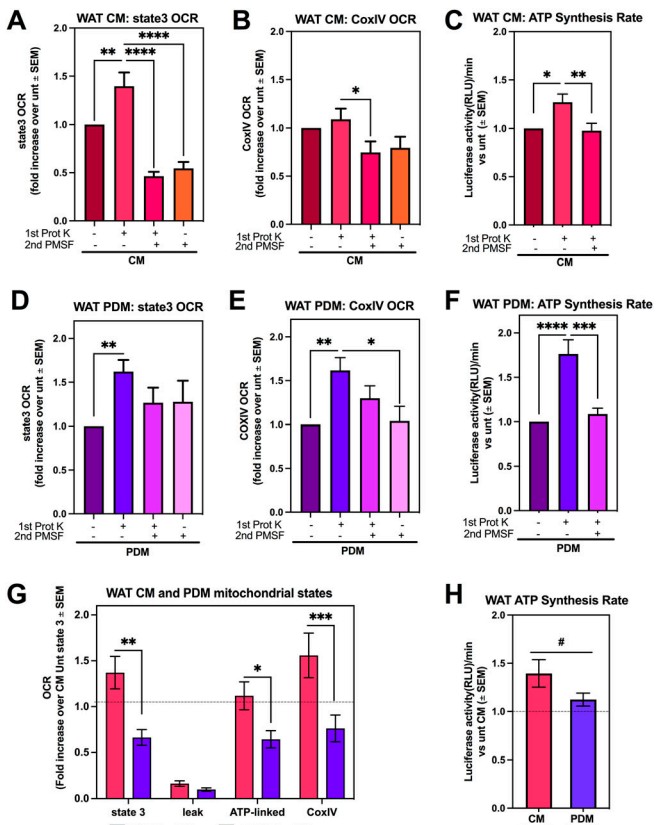

Figure 2. **Proteinase K treatment enables the isolation of functional peridroplet mitochondria (PDM) from white adipose tissue (WAT).**
Assessing mitochondrial function in WAT after isolation with centrifugation alone, Prot K, Prot K + PMSF, or PMSF alone. **(A, B, C)** WAT CM respirometry using pyruvate + malate as the substrate. Data were normalized to untreated CM for each individual experiment. **(A)** State 3 oxygen consumption rate (OCR) of WAT CM from n = 16–20 independent isolation experiments each with one to two biological replicates. **(B)** CoxIV OCR after injection of N,N,N′,N′-Tetramethyl-p-phenylenediamine and ascorbate of WAT CM from 16–20 independent isolations. **(C)** Quantification of ATP synthesis activity in WAT CM from six individual isolations. ATP synthesis rates were determined by the rate of luminescence gain. **(D, E, F)** WAT PDM respirometry using pyruvate + malate as the substrate. Data were normalized to untreated PDM for each individual experiment. **(D)** State 3 OCR of WAT PDM and **(E)** CoxIV OCR of WAT PDM after injection of N,N,N′,N′-Tetramethyl-p-phenylenediamine and ascorbate from 15–20 independent isolations. **(F)** Quantification of ATP synthase activity in WAT PDM from six individual isolations. **(G)** Assessing mitochondrial function in WAT CM and PDM with Prot K from 13–15 independent isolations. Both CM + Prot K and PDM + Prot K were normalized to untreated CM for each individual experiment. **(H)** Quantification of ATP synthesis activity using pyruvate + malate as the substrate from WAT CM and PDM treated with Prot K from 13–15 independent isolations. *$P < 0.05$, **$P < 0.01$, ***$P < 0.001$, ****$P < 0.0001$ by one-way ANOVA. #$P < 0.05$ by $t$ test. All data presented as mean ± SEM.

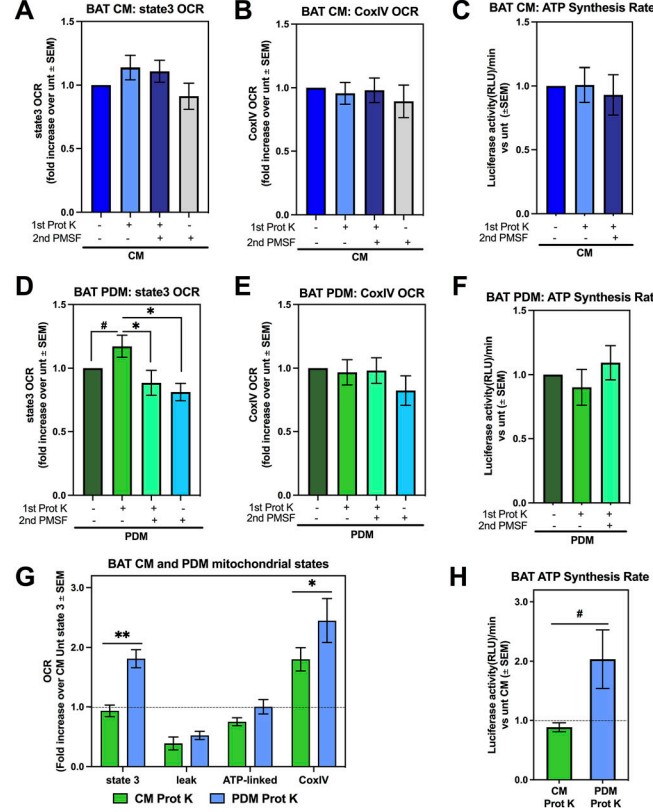

Figure 3. **Isolation of peridroplet mitochondria (PDM) from brown adipose tissue (BAT) with Proteinase K does not decrease mitochondrial respiratory capacity.**
Assessing mitochondrial function in BAT after isolation with regular isolation protocol, Prot K, Prot K + PMSF, or PMSF alone. **(A, B, C)** BAT CM respirometry using pyruvate + malate as the substrate. Data were normalized to untreated CM for each individual experiment. **(A, B)** State 3 oxygen consumption rate (OCR) of BAT CM (A) and (B) CoxIV OCR after injection of N,N,N′,N′-Tetramethyl-p-phenylenediamine and ascorbate of BAT CM from 6–13 independent isolation experiments. **(C)** Quantification of ATP synthesis activity in BAT CM from n = 5 individual experiments. ATP synthesis rates were determined by the rate of luminescence gain. **(D, E, F)** BAT PDM respirometry using pyruvate+malate as the substrate. Data were normalized to untreated PDM for each individual experiment. **(D, E)** State 3 OCR (D) of BAT PDM and (E) CoxIV OCR after N,N,N′,N′-Tetramethyl-p-phenylenediamine and ascorbate injection of BAT PDM from 6–13 independent isolation experiments. **(F)** Quantification of ATP synthase activity in BAT PDM from n = 5 individual experiments. **(G)** Assessing mitochondrial function in BAT CM and PDM with Prot K from n = 9–12 independent isolations. Both CM + Prot K and PDM + Prot K were normalized to untreated CM for each individual experiment. **(H)** Quantification of ATP synthesis activity using pyruvate + malate as the substrate from BAT CM and PDM treated with Prot K from n = 3–4 independent isolations. *$P < 0.05$, **$P < 0.01$ by one-way ANOVA. #$P < 0.05$ by $t$ test. All data presented as mean ± SEM.

new data show that Prot K treatment did not induce mild damage in the outer membrane to cause a small leak in cytochrome c.

Our protocol allowed us to compare the respiratory capacity between CM and PDM in WAT, which revealed remarkable differences between PDM and CM in BAT. PDM showed lower respiratory capacity than CM in WAT, as demonstrated by decreased state 3 and CoxIV-driven respiration (Fig 2G). These results might suggest that the demand for CM pyruvate oxidation in WAT can be higher than in BAT, as CM in BAT are specialized in fatty acid oxidation (Benador et al, 2018).

Confirming the decrease in state 3 respiration fueled by pyruvate, WAT PDM also showed lower ATP-synthesis rates than CM (Fig 2H).

We next determined whether Prot K treatment altered the function of PDM isolated from BAT, and preserved the previously published distinct bioenergetics and proteomic makeup of BAT PDM (Benador et al, 2018). First, we found that, as in WAT, using Prot K to isolate CM from BAT did not decrease state 3 CoxIV-driven respiration or ATP synthesis rates (Fig 3A–C). Similarly, Prot K treatment did not decrease the respiratory function of PDM fractions from BAT (Fig 3D). Indeed, Prot K treatment even increased

state 3 respiration in PDM fractions from BAT, as observed in WAT. We also found that PMSF treatment by itself decreased state 3 respiration in PDM. No significant differences were observed in the measures of CoxIV-driven respiration or ATP-linked respiration in BAT PDM, although PMSF treatment trended towards decreased CoxIV respiration (Fig 3E and F).

The side-by-side comparison of BAT PDM and CM function revealed that Prot K treatment preserved the unique characteristics of BAT PDM, which we previously published (Benador et al, 2018). BAT PDM fueled by pyruvate and malate show higher state 3 and CoxIV-stimulated maximal respiration than CM (Fig 3G). Furthermore, we were able to reproduce the previous observation that BAT PDM have higher ATP synthesis rates than BAT CM (Fig 3H) (Benador et al, 2018). Overall, these data suggest that Prot K-assisted isolation does not inhibit mitochondrial respiratory function and that PMSF treatments are only needed to analyze the mitochondrial proteome for further characterization of mitochondrial subpopulations (Forner et al, 2009; Benador et al, 2018; Mirza et al, 2021; Najt et al, 2023).

### PDM isolation from lipid droplets with different diameters reveals that lipid droplet size sets PDM functional heterogeneity

Previous studies have shown that LDs vary in size within individual cells and that these differently sized LDs can have distinct metabolic functions (Herms et al, 2013; Zhang et al, 2016; Olzmann & Carvalho, 2019). Furthermore, changes in lipid droplet size are a response to cycles of nutrient scarcity and overload, as lipid droplets buffer NEFAs to protect from lipotoxicity (Herms et al, 2013; Rambold et al, 2015; Nguyen et al, 2017; Hariri et al, 2018, 2019; Olzmann & Carvalho, 2019; Renne & Hariri, 2021).

Electron microscopy images of WAT and BAT show that although WAT has larger sized lipid droplets, it has fewer mitochondria per total lipid droplet perimeter per cell. In contrast, BAT contains smaller sized lipid droplets and has more mitochondria per total lipid perimeter (Cushman, 1970; Cinti, 2000, 2001, 2007, 2018). We hypothesized that small LDs recruited more mitochondria, despite showing less perimeter of contact area. To decipher if the recruitment of mitochondria to LDs is determined by the size of the LD, and if this regulation is tissue-specific, we developed an approach to obtain separated LD fractions that differed in the average size of their LDs by performing two consecutive centrifugations (Brasaemle & Wolins, 2016; Zhang et al, 2016). The approach required us to be as gentle as possible to yield functional isolated mitochondria.

The first centrifugation at 500$g$, yields a fat layer that contains large LDs (LG-LD). This was followed by a second centrifugation step at 2,000$g$, resulting in a fat layer that consists of smaller LDs (SM-LD) (Fig 4A). To confirm that these centrifugation steps indeed separated two fractions with differently sized LDs, we stained the LG-LD and SM-LD fractions with BD493 and used confocal microscopy to visualize the LD size (Fig 4B and C). More specifically, we measured the perimeter and size distribution by surface area of each individual LD imaged in the large and small LD fractions isolated from WAT and BAT. We found that the difference in area between the large and small fractions is more pronounced in the WAT, with LG-LD having an average area of 328.6, $\mu m^2$, whereas SM-LD had an average area of 1.9 $\mu m^2$ (Fig 4B and D). In contrast, the range

between the LG and SM-LD average area is smaller in the BAT: the average area of LG-LD was 40.2 $\mu m^2$, whereas 3.4 $\mu m^2$ was the average of SM-LD (Fig 4C and E). Accordingly, the average LD perimeter of the LG-LD in WAT is 37.6, and 2.5 $\mu m$ in SM-LD (Fig 4F). In BAT, the average LD perimeter in the LG-LD fraction is 18.6, and 5.4 $\mu m$ in SM-LD (Fig 4G). Of note, the large lipid droplet fraction isolated from the WAT still contains a significant amount of smaller LDs (Fig 4F). Moreover, the lipid droplet sizes we observed in the isolated fat layers are consistent with what has been reported in vivo in previous studies (Swift et al, 2017).

To measure whether small lipid droplets had more mitochondria bound to them, we compared the amount of PDM between LG-LD and SM-LD fractions, by quantifying the ratio of MTDR/BD493 fluorescence using our published plate reader assay of PDM quantification (Acín-Perez et al, 2021). Our data show that the SM-LD fraction contains more mitochondria per lipid content when compared with the LG-LD fraction in WAT (Fig 4H). However, we found that in BAT the LG-LD fraction has more mitochondria attached to the LDs when compared with the SM-LD fraction (Fig 4I).

### Mitochondria attached to small and large lipid droplets have different respiratory capacities in WAT and BAT

Separation of lipid droplets by size from mouse BAT revealed diverse protein profiles between the subpopulations of lipid droplets, indicating different interactions with mitochondria and the ER (Zhang et al, 2016). In intact primary brown adipocytes, we observed heterogeneity in PDM function and composition; some PDM showed higher ATP synthase content and membrane potential than others (Benador et al, 2018). Here, we have observed that PDM from WAT have lower ATP-synthesizing respiration and CoxIV activity than CM, which is the opposite behavior of BAT PDM (Figs 2 and 3). We find that WAT harbors very large lipid droplets (>100 $\mu m$ perimeter) that do not exist in BAT from mice housed at RT and fed a standard chow diet (Fig 4E). Based on these observations, we sought to test our hypothesis that LD size might determine the intracellular and tissue-specific heterogeneity of PDM function.

To test this hypothesis, we stripped PDM from SM-LD and LG-LD fractions and analyzed them separately by respirometry. In WAT, PDM isolated from SM-LD had significantly higher state 3 OCR when respiring under pyruvate and malate, and higher CoxIV-driven respiration per mitochondria when compared with PDM isolated from LG-LD (Fig 5A). Under palmitoyl carnitine as the oxidative fuel, the differences in mitochondrial respiration disappeared between WAT PDM isolated from LG versus SM-LD (Fig 5B).

Interestingly, we find that PDM isolated from BAT LG-LDs show higher state 3 respiration, proton leak-driven respiration, and ATP-synthesizing respiration when compared with PDM from SM-LDs under pyruvate malate (Fig 5C). In marked contrast to WAT, CoxIV-driven respiration in BAT was similar in PDM from LG-LD and SM-LD fractions. Furthermore, BAT PDM from LG-LD fractions preserved increased state 3, leak, and ATP-synthesizing respiration, in addition to increased CoxIV when palmitoyl–carnitine was provided as a fuel (Fig 5D).

To validate that the differences in respirometry between the two fractions in BAT and WAT are because of bioenergetic differences rather than differences in fraction purity, we measured the content

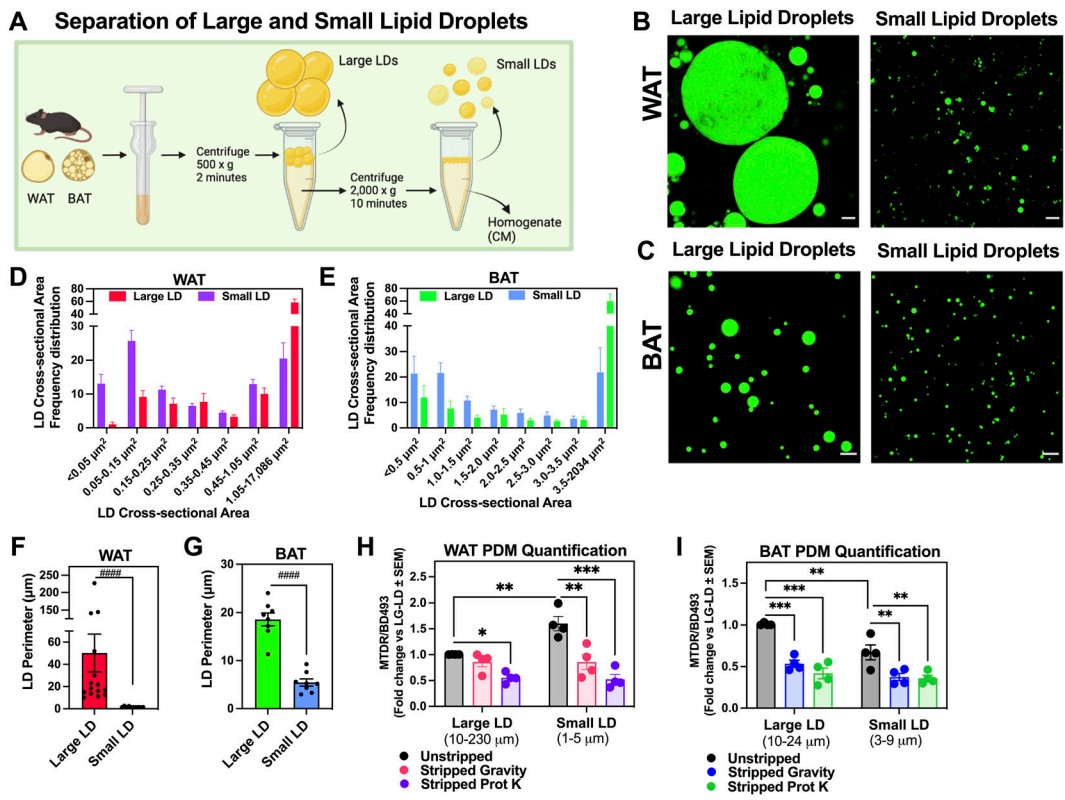

**Figure 4. Proteinase K treatment enables the isolation of mitochondria attached to small lipid droplets that cannot be detached by centrifugation.**
**(A)** Schematic representation of the isolation protocol for small and large LDs. Tissue was homogenized and first centrifuged at 500$g$ to create a fat layer containing large LDs (LG-LDs). After removing the first fat layer, homogenates were centrifuged at 2,000$g$ to create a second fat layer containing small LDs (SM-LDs). **(B, C)** Super-resolution confocal microscopy of fat layers containing LG-LDs and SM-LDs isolated from white adipose tissue (WAT) (B) and brown adipose tissue (BAT) (C). The fat layers were stained with BD493. Scale bar = 10 microns. **(D, E)** Examining LD size distribution in fat layers isolated from WAT (D) and BAT (E) separated by differential centrifugation. N = 4–6 individual experiments with n > 2 images analyzed per experiment. **(D)** In WAT, the data were grouped into bins with a range of 0.1 starting at 0. The last group contained the frequency of LD area between 1.05 and 17,086 $\mu m^2$. **(E)** In BAT, the bin range was 0.5 and started at 0. The last group contained the frequency of LD area between 3.5 and 2,034 $\mu m^2$. **(F, G)** Quantification of the perimeter of lipid droplets found in the large LD and the small LD preparations isolated from WAT (F) and BAT (G) fat layers. N = 4–6 individual experiments with n > 2 images analyzed per individual experiment. Each point on the graph represents the average LD perimeter from a single image. **(H, I)** Assessing the amount of peridroplet mitochondria (PDM) associated with large versus small LDs. PDM quantification from WAT (H) and BAT (I) using the fluorescence plate reader assay in n = 4 individual experiments. Mitochondria were stained with Mitotracker DeepRed and LDs with BD493. The ratio of Mitotracker DeepRed/BD493 was quantified as a measure of PDM abundance. *$P < 0.05$, **$P < 0.01$, ***$P < 0.001$ by one-way ANOVA, ####$P < 0.0001$ by nonparametric Kolomogrov–Smirnof analysis. All data presented as mean ± SEM. Illustrations were created in BioRender.

of mitochondrial proteins (VDAC and OXPHOS subunits) per total protein of PDM fraction by Western blot. PDM isolated from WAT SM-LD fractions had a significantly higher content of VDAC. In addition, WAT PDM isolated from SM-LD had increased levels of respiratory complexes as determined by the expression of representative proteins in four out of five OXPHOS complexes (NDUFB8, SDHB, UQCRC2, and MTC01) (Fig 5E and F). Therefore the respirometry data for WAT (Fig 5A and B) were normalized by VDAC, as a marker for total mitochondrial content (Camara et al, 2017). On the other hand, BAT PDM isolated from large and small LDs did not show differences in VDAC or OXPHOS subunit protein content and the respirometry did not need to be re-normalized (Fig 5G and H).

Together these data suggest that in both WAT and BAT, there are separate populations of PDM with unique bioenergetic functions, which may be related to their distinct subcellular localization and heterogeneity of LD function.

## Discussion

Here, we describe the development of the first method allowing for the isolation of PDM from WAT. A study by Benador et al previously described the isolation and characterization of PDM from BAT using high-speed centrifugation (Benador et al, 2018). However, we found that centrifugation was not an efficient approach for the isolation of PDM from WAT. The novelty of our new method is the treatment of lipid droplet fractions, containing PDM, with a protease. This step was inspired by previous studies supporting that protein–protein interactions and protein–lipid interactions are key tethering mechanisms (Stone et al, 2009; Wang et al, 2011; Boutant et al, 2017; Freyre et al, 2019). We used Prot K, a serine endopeptidase that remains active in the presence of different detergents and temperature ranges, and digests a wide variety of proteins. The dependence upon Prot K to isolate sufficient WAT PDM supports the notion that the interaction of

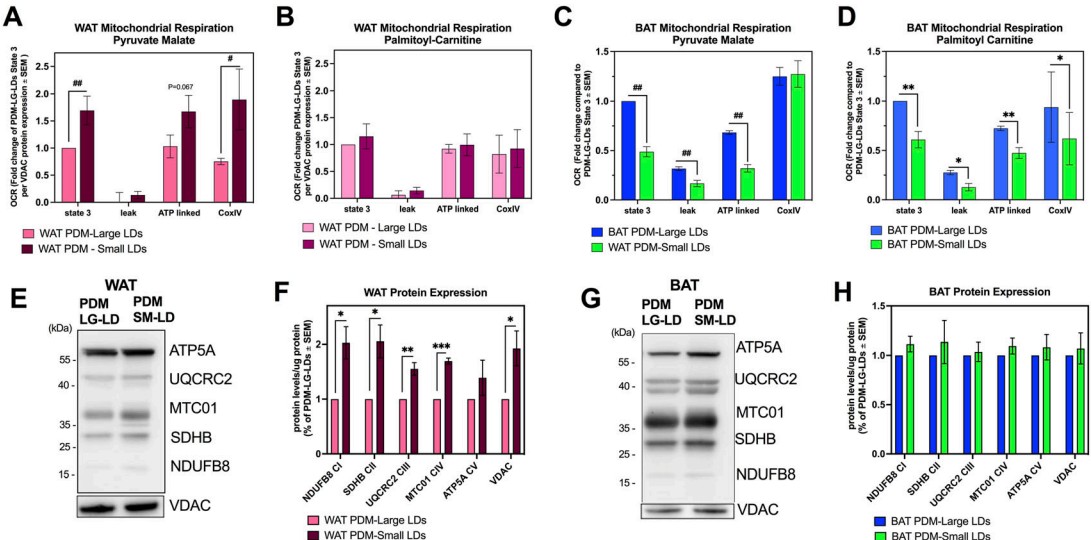

**Figure 5. Peridroplet mitochondria (PDM) isolated from large and small lipid droplets have unique characteristics.**
Assessing mitochondrial function in PDM isolated from LG- and SM-LDs from white adipose tissue (WAT) and brown adipose tissue (BAT). All mitochondria were isolated from WAT and BAT using the Prot K protocol. **(A, B)** Quantification of state 3, mitochondrial proton leak, ATP-linked, and CoxIV oxygen consumption rate using pyruvate malate (A) as substrate or palmitoyl-carnitine + malate (B) as substrate in PDM from LG- and SM-LDs in WAT. Four to six individual isolations were normalized to VDAC protein content for each experiment as a measure of mitochondrial content. **(C, D)** Assessing mitochondrial function in PDM isolated from LG- and SM-LDs in BAT. Quantification of state 3, mitochondrial proton leak, ATP-linked, and CoxIV oxygen consumption rate using pyruvate + malate (C) as substrate or palmitoyl-carnitine + malate (D) as substrate from four to six individual isolations. **(E, F)** Representative Western blots of mitochondrial proteins from mitochondria isolated from LG- and SM-LDs from WAT and (F) quantification of the protein data from n = 3–5 individual isolations. **(G, H)** Representative Western blots of mitochondrial proteins from mitochondria isolated from LG- and SM-LDs from BAT and (H) quantification of the protein data from n = 3–5 individual isolations. *$P < 0.05$, **$P < 0.01$, ***$P < 0.001$ by $t$ test and #$P < 0.05$, ##$P < 0.01$ by nonparametric Kolomogrov–Smirnof analysis. All data presented as mean ± SEM.

mitochondria with lipid droplets in WAT might be more resilient to mechanical forces than in BAT. Moreover, the difference in strength between mitochondria and LDs can be mediated either by a different composition of protein or lipid tethers, as supported by differences in PLIN5 and MIGA2 between BAT and WAT, and/or by a larger number of other uncharacterized tethers shared between WAT and BAT (Wolins et al, 2006; Freyre et al, 2019).

In the current protocol, we find that the use of Prot K results in PDM and CM fractions with higher respiratory capacity when compared with those obtained by high-speed centrifugation alone. We propose that this increase can be explained by Prot K degrading few contaminant components that decrease mitochondrial function. This hypothesis is consistent with several studies in which the isolation of interfibrillar mitochondria from both cardiac and skeletal muscle with the protease Nagarse resulted in increased mitochondrial function and energetic coupling (Kras et al, 2016; Koncsos et al, 2018).

We show that Prot K does not digest respiratory complexes in intact mitochondria. However, Prot K inactivation by PMSF is required to prevent degradation in freeze–thawed PDM. Membrane-disrupting processes enable Prot K to access inner membrane proteins and Prot K itself is resistant to detergents and freeze–thawing procedures. Thus, the addition of PMSF is needed to analyze the protein composition of PDM, as time constraints prevented protein analyses on the same day of PDM isolation.

We find that respirometry of PDM isolated with Prot K in BAT recapitulated our previous findings: higher maximal ATP synthesis fueled by pyruvate oxidation, when compared with cytosolic mitochondria (Benador et al, 2018). In marked contrast, WAT PDM have lower respiratory capacity and ATP synthesis rates as compared with WAT CM. Indeed, BAT CM are specialized in oxidizing fatty acids for thermogenesis, a function absent in CM from mature white adipocytes. These data suggest that the functional specialization of PDM in WAT may not be the same as in BAT, and this difference between WAT and BAT may be related to the specific capacity of BAT to perform thermogenesis and fatty acid oxidation. Future studies will determine the function of WAT PDM in the esterification of NEFA, versus de novo lipogenesis and LD homeostasis.

Differences in the size and location of intracellular lipid droplets, and the architecture of their contact sites, are thought to be key determinants of LD function (Renne & Hariri, 2021). These differences in size and location vary depending on the type of cell, nutrient availability, and metabolic state (Thiam & Beller, 2017). The diversity in the multifunctional lipid droplet populations is reflected by the recent characterization of their heterogeneous proteomes and lipid compositions (Martin et al, 2005; Hsieh et al, 2012; Larsson et al, 2012; Wilfling et al, 2013; Thiam & Beller, 2017; Qian et al, 2023). A novel integral endoplasmic reticulum membrane protein expressed selectively in BAT, CLSTN3β, was recently shown to regulate LD form and function. Forced expression of CLSTN3β, in WAT was shown to induce multilocularity and promote fatty acid oxidation, as observed in thermogenic adipocytes (Qian et al, 2023). Moreover, the expression of lipid-droplet associated proteins including ATGL, Plin5, and several CIDE proteins known to regulate BAT multilocular thermogenic phenotype, were induced in WAT upon cold stimulation, highlighting the relationship between lipid droplet size and mitochondrial oxidative function (Barneda et al, 2013).

Our data demonstrate that in WAT, PDM attached to smaller LDs have higher respiratory capacity under pyruvate and malate. In contrast, PDM attached to larger LDs in BAT are the ones that have higher ATP synthesizing capacity, which is the opposite of what we observed in WAT. The discrepancy between BAT and WAT could be explained by the fact that BAT has a larger capacity for de novo lipogenesis and TAG turnover, which is dependent on ATP-synthesizing respiration. Moreover, the study by Benador et al previously showed that PDM support LD expansion by providing ATP for fatty acid esterification into triglycerides, which would explain why PDM attached to larger LDs have higher ATP-synthesizing capacity (Benador et al, 2018). Indeed, BAT is a more dynamic tissue in terms of lipid metabolism than WAT, which is reflected in its multilocular LD morphology and by the constant cycles of lipolysis for $\beta$-oxidation and re-esterification.

On the other hand, WAT shows high rates of de novo lipogenesis when its adipocytes are differentiating, which requires a large amount of ATP as their LDs expand. Morphologically, these differentiating white adipocytes have multilocular LDs, which are very similar to brown adipocytes. PDM from small LDs in WAT have increased pyruvate oxidation and ATP-synthase capacity, supporting the idea that PDM from small LDs provide ATP necessary for lipid droplet expansion. This is also supported by the report that white adipocytes increase mitochondrial biogenesis during differentiation to meet the increased energy demand of differentiation, as cells and their lipid droplets mature (Wilson-Fritch et al, 2003; De Pauw et al, 2009).

We show that fat oxidation capacity is higher in PDM isolated from large lipid droplets in BAT when compared with smaller BAT lipid droplets. On the other hand, we did not see differences in fat oxidation capacity of WAT PDM from large and small lipid droplets. Remarkably, the large lipid droplet fraction in BAT contains lipid droplets of the same size as the small lipid droplet fraction in WAT. This may reflect a relationship between fat utilization efficiency and LD size, where fat oxidation capacity is maximal at a certain lipid droplet size and does not further increase after a certain threshold of size.

From the current experiments, we cannot definitively say whether the higher respiratory capacity induces PDM to bind preferentially to smaller LDs or if the higher respiratory capacity of PDMs fueling the oxidation of fatty acids leads to the observation of more active PDM on smaller LDs in WAT. However, fat oxidation is minimal in mature WAT. This suggests that when mitochondria interact with small LDs in WAT, they are not oxidizing these fatty acids. At the same time, as previously described, the small LDs in WAT are the ones that can expand and are thus more active in esterifying and breaking lipids. Therefore, we suggest that in WAT, the interaction of PDM with small LDs and the higher capacity for ATP synthesis supports a higher activity of TG esterification in small LDs compared with large LDs.

Consistent with our data, it was recently shown that white pre-adipocytes lacking MIGA2, the protein linker that binds mitochondria to LDs in WAT, had reduced adipocyte differentiation, decreased LD abundance, and diminished TAG synthesis (Freyre et al, 2019). The increased surface area of small multilocular LDs can promote increased lipolysis and the subsequent release of NEFA (Nishimoto & Tamori, 2017). Furthermore, in MIGA-2 knockout pre-adipocytes, radio-labeled glucose is not enriched in TAGs, suggesting that PDM association to LDs is essential for the expansion of small lipid droplets through de novo lipogenesis and pyruvate oxidation in WAT (Freyre et al, 2019). Once white adipocytes have fully matured and increased their LD size to form a single unilocular LD, they reduce their need for ATP to actively perform TAG synthesis, which is one of the roles of PDM described in BAT.

When PDM from large and small LDs are isolated together, as they were in our initial experiments, the functional differences between the PDM from different populations are averaged together, concealing their unique properties. The new isolation protocol presented here allows us to explore open questions related to mitochondria–LD interaction and will help us better understand PDM function within individual cells and between different adipose tissue depots. Overall, we show data supporting the existence of at least two different populations of PDM, distinguished by their association with LDs of varying sizes. The differences in mitochondrial function observed between the subpopulations potentially reflect local LD environments with specific needs. We show that to isolate PDM from WAT, Prot K treatment is needed regardless of the size of the LD that is being used for the preparation. As the yield of PDM isolated from SM-LD and LG-LD by centrifugation is similar, we can conclude that the strength of PDM attachment to different size LDs is similar in BAT. We do, however, observe more PDM attached to smaller LDs in WAT, and the opposite in BAT. This discrepancy between BAT and WAT supports that other tissue-specific factors beyond lipid droplet size determine mitochondria–LD interaction.

Understanding what controls the diversity of lipid droplet biology can help us better understand how and why such diversity exists, and shed new light on the role of LD biology in determining the function of PDM.

A limitation of the approach presented here is in the utilization of crude isolated mitochondrial fractions for the respirometry studies. Crude isolated mitochondria may include diverse levels of contamination by other organelles. A caveat to alternative purification methods which may include ultracentrifugation is that they are time-consuming and, although they result in purer mitochondrial fractions, they compromise mitochondrial integrity and impair their function. Biochemical studies, for example, might require different isolation protocols depending on the main aim of the study. Another limitation of our approach is our incomplete understanding of the enzymatic impact of Prot K on mitochondria. Although we show here that mitochondria exposed to Prot K have comparable and even higher state 3 and ATP-synthesizing respiration, this does not preclude that regulatory systems that are dependent on outer membrane proteins are disabled by the Prot K. Further studies would be required to better understand the effect of Prot K on respiration and individual outer membrane proteins.

In this study, we developed a new method to isolate PDM from WAT and improved the isolation of PDM from BAT by adding proteolytic treatment. We demonstrate that the addition of Prot K to the PDM isolation protocol does not impair mitochondrial OCR and ATP synthesis capacity in both CM and PDM. Using our new protocol to isolate PDM from WAT, we show that WAT PDM have a lower respiratory capacity than WAT CM, in contrast to our previously published findings on PDM in BAT. We also show that different-sized lipid droplets in both BAT and WAT are associated with unique populations of PDM, highlighting the heterogeneity in PDM function. These data suggest PDMs have distinct roles in maintaining LD

homeostasis, which is determined by the association with LDs and the local cellular needs.

# Materials and Methods

### Mice

Mitochondria were isolated from the interscapular BAT and both left and right epididymal white adipose fat pads from 12-wk-old male C57BL6/J mice (Jackson lab). Animals were fed standard chow (mouse diet 9F; PMI Nutrition International) and maintained under controlled conditions (19–22°C and a 14:10 h light–dark cycle) until euthanasia by isoflurane. All animal procedures were performed in accordance with the Guide for Care and Use of Laboratory Animals of the NIH and were approved by the Animal Subjects Committee of the University of California, Los Angeles Institutional Guidelines for Animal Care.

### PDM isolation

PDM were isolated as previously described in detail (Benador et al, 2018; Ngo et al, 2021) with some modifications. BAT was isolated from the inter-scapular BAT depot, and WAT was isolated from both perigonadal fat pads, also known as the epididymal fat from each mouse. BAT was homogenized using a glass–teflon dounce homogenizer, and WAT was homogenized using a glass–glass dounce homogenizer in Sucrose-HEPES-EGTA buffer supplemented with BSA (SHE + BSA; 250 mM sucrose, 5 mM HEPES, 2 mM EGTA, 1% fatty acid-free BSA, pH 7.2). Homogenate was split into four equal parts to test protocol optimizations side-by-side. Homogenates were centrifuged at 1,000$g$ for 10 min at 4°C. The supernatant was poured into a new tube and the fat layer was scraped into a second tube and resuspended in SHE + BSA buffer. Fat layers and supernatant were left either untreated or treated with Prot K at 2 $\mu$g/ml for 15 min (Prot K, 25530-049 20 mg/ml; Invitrogen). All fat layers were incubated for 15 min at 4°C under constant rotation. For protocol including PMSF, 2 mM PMSF was added for an additional 20 min of incubation on ice while inverting the samples every 2 min (PMSF, 78830; Sigma-Aldrich). Then, all the samples were centrifuged again at 10,000$g$ for 10 min at 4°C. The pellets were then resuspended in SHE + BSA and centrifuged with the same settings once more. The pellets were then resuspended in SHE without BSA and again centrifuged with the same settings. Final pellets were resuspended in SHE without BSA and protein concentration was determined by BCA (Thermo Fisher Scientific).

### Isolation of peridroplet mitochondria from large and small lipid droplets

Fat tissue was homogenized as described above for the PDM isolation. Homogenates were first sieved through a 630-$\mu$m mesh (Genesee Scientific), and transferred into ice-cold tubes. Homogenates were then centrifuged at 500$g$ for 3 min at 4°C which created a fat layer made up of the larger lipid droplets (LG-LD). The supernatant was carefully poured into a new tube and centrifuged at 2,000$g$ for 10 min at 4°C which created a second fat layer consisting of smaller lipid droplets (SM-LD). The supernatant which contained the cytoplasmic mitochondria was transferred into a new tube by carefully pipetting underneath the fat layer. LG-LDs and SM-LDs were resuspended in SHE + BSA. LG-LDs were centrifuged again at 500$g$ for 5 min at 4°C and SM-LGs were centrifuged at 2,000$g$ for 10 min at 4°C. Both large and small LDs were resuspended in 1 ml SHE + BSA and incubated with Prot K as described above. All fractions including the supernatant containing the CM were centrifuged at 10,000$g$ for 10 min at 4°C to pellet the mitochondria. Mitochondrial pellets were resuspended in SHE without BSA and spun one more time with the same conditions. Mitochondrial pellets were resuspended in SHE without BSA and protein concentrations were determined by BCA (Thermo Fisher Scientific).

### Isolated mitochondria respirometry

Respirometry in isolated mitochondria was performed as previously described in detail (Ngo et al, 2021). Briefly, isolated mitochondria were resuspended in a mitochondrial assay buffer (MAS; 100 mM KCl, 10 mM KH$_2$PO$_4$, 2 mM MgCl$_2$, 5 mM HEPES, 1 mM EGTA, 0.1% BSA, 1 mM GDP, pH 7.2) and kept on ice. Two micrograms per well were loaded into Seahorse XF96 microplate in 20 $\mu$l volume containing substrates. The loaded plate was centrifuged at 2,000$g$ for 5 min at 4°C and an additional 130 $\mu$l of MAS buffer + substrate was added to each well. Substrate concentrations in the well were as follows: (i) 5 mM pyruvate + 5 mM malate + 4 mM ADP or (ii) 2 mM malate + 4 mM ADP. For pyruvate plus malate-dependent respiration; oligomycin was injected at port A (3.5 $\mu$M), TMPD + ascorbic acid (0.5 + 1 mM) at port B and azide (50 mM) at port C. For palmitoyl–carnitine dependent respiration; 5 mM malate + 4 mM ADP + 40 $\mu$M palmitoyl–carnitine was injected at port A, oligomycin was injected at port B (3.5 $\mu$M), TMPD + ascorbic acid (0.5 + 1 mM) at port C and azide (50 mM) at port D. Mix and measure times were 0.5 and 4 min, respectively. A 2-min wait time was included for oligomycin-resistant respiration measurements.

To calculate the effects of Prot K on OCR, we measured state 3 or the capacity of coupled mitochondria to produce ATP in the presence of substrates and ADP, oligomycin-resistant leak and ATP-linked respiration, Complex IV-driven respiration, and non-mitochondrial oxygen consumption.

Fold changes in respiration were calculated from raw OCR values obtained in each experimental group, where respiration of treated mitochondria isolated on the same day and analyzed on the same plate were normalized to the untreated mitochondria to establish the effect of PK and PMSF treatment. The individual fold changes over untreated mitochondria obtained from independent experiments were then averaged, and represented in bar graphs. For each substrate (pyruvate malate or palmitoyl carnitine) and mitochondrial state (i.e., State 3, leak, ATP-linked, CoxIV), we calculated the fold change of each mitochondrial state between the mitochondria treated with PK and/or PMSF and compared it with the untreated CM or PDM of the group. For example for State 3, we set the CM untreated equal to 1 by dividing by the raw OCR value and then compared the raw OCR values of the 3 other groups with the CM untreated to calculate the fold change. For the comparison between CM Prot K and PDM Prot K as in Figs 2G and 3G, the fold

changes were calculated from the raw OCR values of each experiment by comparing the OCR values of each state with the CM untreated state 3, which was set to 1.

## Peridroplet mitochondria quantification

### Plate reader assay

The fat layer and stripped fat layer were incubated in MAS buffer containing MTDR from Invitrogen M22426 (500 nM, final) and BODIPY493/503 from Invitrogen D3922 (BD493, 1 $\mu$M, final) for 10 min at 37°C. Dye was removed by centrifuging the samples at 1,000$g$ for 10 min and removing the infranatant. The stained fat layer or stripped fat layer was resuspended in 100 $\mu$l of MAS and fluorescence was measured in a clear-bottom black 96-well plate (Corning, NY). MTDR was excited at 625 nm and its emission was recorded at 670 nm. BD493 was excited at 488 nm laser and its emission recorded at 500–550 nm. MTDR/BD493 was calculated for the fat cake and the stripped fat cake. For PDM quantification, we calculated the difference of the MTDR/BD493 between the fat cake and the stripped fat cake.

### ATP synthesis assay

10 $\mu$g of isolated mitochondria were resuspended in 10 $\mu$l MAS buffer containing 5 mM pyruvate + 5 mM malate + 3.5 mM ADP and plated onto a clear-bottom black 96-well plate (Corning, NY). Luciferin–luciferase mix was added to the mitochondria. Luminescent counts were integrated over 0.5 s at 10 s intervals separated by 0.5 s orbital shaking on Spark M10 microplate reader (Tecan). The linear rate of luminescence increase was calculated to determine ATP synthesis rate.

### Protein gel electrophoresis and immunoblotting

5–15 mg of isolated mitochondrial protein or fat layer was resuspended in NuPAGE LDS Sample Buffer containing $\beta$-mercaptoethanol (Thermo Fisher Scientific). Samples were then loaded into 4–12% Bis–Tris precast gels (Thermo Fisher Scientific) and electrophoresed in constant voltage at 60 V for 30 min (to clear stacking) and 140 V for 60 min. Proteins were transferred to methanol-activated Immuno-Blot PVDF Membrane (Bio-Rad) in 30 V constant voltage for 1 h at 4°C. Blots were incubated overnight with a primary antibody diluted in PBST (phosphate buffered saline with 1 ml/liter Tween-20/PBS) + 5% BSA (Thermo Fisher Scientific) at 4°C. The next day, blots were washed in PBST and incubated with fluorescent or HRP secondary antibodies, diluted in PBST+ 5% BSA for 1 h at RT. Proteins were detected using the following antibodies: MTCO1 antibody (1D6E1A8) (ab14705), Total Rodent OXPHOS Cocktail (ab110413), VDAC (ab15895) all from Abcam. ATP5A1 Monoclonal Antibody (15H4C4) (43-9800; Thermo Fisher Scientific), NDUFFB8 monoclonal antibody (20E9DH10C12) (459210; Thermo Fisher Scientific), SDHA Monoclonal Antibody (2E3GC12FB2AE2) (459200; Thermo Fisher Scientific) from Thermo Fisher Scientific. UQCRC2 Rabbit Polyclonal Antibody (14742-1-AP) from Proteintech, TOMM20 monoclonal antibody clone 4F3 (WH0009804M1-100UG) Sigma-Aldrich and VDAC (D73D12) Rabbit mAb Cell signaling Technology 4661T. Secondary antibodies used were goat anti-Mouse IgG secondary antibody, Alexa Fluor 660

conjugate (Thermo Fisher Scientific), goat anti-rabbit IgG secondary antibody DyLight 800 (Thermo Fisher Scientific), donkey anti-mouse IgG (H + L) Highly Cross Adsorbed Secondary Antibody, Alexa Fluor 488 conjugate (Thermo Fisher Scientific), anti-rabbit IgG, HRP-linked antibody (7074; Cell Signaling Technology), anti-mouse IgG, HRP-linked antibody (7076; Cell Signaling Technology). Blots were imaged on the ChemiDoc MP imaging system (Bio-Rad Laboratories). Band densitometry was quantified using FIJI (ImageJ, NIH).

## Fluorescence microscopy

### Confocal microscope

All imaging was performed on Zeiss LSM880. Super-resolution imaging was performed with 63x and 40x Apochromat oil-immersion lens and Airyscan super-resolution detector.

### Fluorophore excitation/emission

All fluorophores were excited on separate tracks to avoid artifacts because of bleed-through emission. BODIPY 493/503 were excited with a 488-nm 25 mW Argon-ion laser and their emission captured through 500–550 nm band-pass filter. MTDR was excited using a 633-nm 5 mW helium–neon laser and its emission was captured through a 645 nm long-pass filter.

### Lipid droplet imaging

Fractions of large and small lipid droplets were stained with BODIPY493/503 (BD493, 1 $\mu$M, final) for 10 min at 37°C. 10–20 $\mu$l of each sample was mixed in a 1:1 ratio with Matrigel and pipetted into a single compartment of a cellview glass bottom four-compartment cell culture dish (#627975; Griener Bio-One). Imaging was performed using a 63x Apochromat oil-immersion lens for lipid droplet layers.

### PDM imaging from fat layers

Pre-stripped fat layer and stripped fat layer were incubated in MAS buffer containing MTDR, (500 nM, final) and BODIPY493/503 (BD493, 1 $\mu$M, final) for 10 min at 37°C. Dye was removed by centrifuging the samples at 1,000$g$ for 10 min and removing the infranatant. The stained fat layer or stripped fat layer was resuspended in 100 $\mu$l of MAS. 20 $\mu$l of each sample was mixed in a 1:1 ratio with Matrigel and pipetted into a single compartment of a Cellvis glass bottom four-compartment cell culture dish (#627975; Griener Bio-One). Imaging was performed using 63x for BAT and both 40x and 63x Apochromat oil-immersion lens for WAT.

## Image analysis

### Lipid droplet size analysis

Fat layers of large and small LDs stained with BD493 were measured by fluorescence detection using AIVIA image analysis software (version 10.5.0; Leica Microsystems). The LDs were detected and segmented by BD493 fluorescence using pixel classifier machine learning that generated a segmentation mask. LD count and measures of perimeter and surface area occupied by BD493 pixels were extracted by AIVIA software from the segmentation data for each image. The average perimeter and surface area for each image were calculated and graphed as a single point.

### Analysis of PDM content from fat layers

Mitochondria–lipid droplet interactions in 2D images of BD493 and MTDR-stained fat layers were analyzed with CellProfiler 2.0 (Kamentsky et al, 2011). LDs were identified based on BD493 staining with size cutoffs of 20–100 microns for WAT LDs and 1–50 microns for BAT LDs. Mitochondrial staining was deblurred by subtraction through a 52-pixel median filter. Subsequently, mitochondrial ROIs of 0.5–5 microns were recognized through an adaptive Otsu thresholding algorithm. Mitochondrial ROIs within 500 nm distance to the LD border were classified as PDM, as adopted from Benador et al (2018), and the percentage of interaction was determined as the length of the perimeter of the LD boundary interacting with mitochondria versus the length of total LD perimeter.

### Image presentation

Image contrast and brightness were not altered in any quantitative image analysis protocols. Brightness and contrast were optimized to properly display representative images in figure panels.

## Statistical analysis

Statistical analyses were performed using GraphPad Prism 5.03 (GraphPad Software Inc.). Data were presented as mean ± SEM for all conditions. Normality of data was checked by the Kolmogorov–Smirnov test with the Dallal–Wilkinson–Lillie for corrected $P$-value. For data with normal distribution, one-way ANOVA with Šídák multiple comparisons test was employed. Individual means were compared using the parametric two-tail $t$ test. For nonparametric data, Kruskal–Wallis with Dunn's multiple comparisons test was used. Comparisons between two groups were assessed by the Kolmogorov–Smirnov test, and when appropriate Wilcoxon matched-pairs rank test. Differences of $P < 0.05$ were considered to be significant. All graphs and statistical analyses were performed using GraphPad Prism 9 (GraphPad Software).

## Supplementary Information

## Acknowledgements

We would like to thank Dr. Ilan Benador, Dr. Dani Dagan, Dr. Jennifer Ngo, Dr. Cristiane Beninca, and Dr. Michael Shum for helpful discussions and advice. Figures created with BioRender.com. This work was supported by the US NIH grants R01DK099618-05 (OS Shirihai), R01CA232056-01 (OS Shirihai), R21AG060456-01 (OS Shirihai), R21AG063373-01 (OS Shirihai). M Liesa was funded by 1R01AA026914-01A1, DGSOM Seed Award and is funded by MCINN/AEI/FEDER PID2021-127278NB-I00.

## Author Contributions

AJ Brownstein: conceptualization, resources, data curation, formal analysis, supervision, funding acquisition, validation, investigation, methodology, and writing—original draft, review, and editing.

M Veliova: conceptualization, data curation, formal analysis, validation, investigation, methodology, and writing—original draft, review, and editing.

R Acin-Perez: conceptualization, supervision, validation, investigation, methodology, and writing—review and editing.

F Villalobos: conceptualization, data curation, supervision, validation, methodology, and writing—review and editing.

A Petcherski: data curation, validation, methodology, and writing—review and editing.

A Tombolato: data curation, validation, methodology, and writing—review and editing.

M Liesa: conceptualization, resources, supervision, funding acquisition, and writing—original draft, review, and editing.

OS Shirihai: resources, supervision, funding acquisition, methodology, and writing—review and editing.

## Conflict of Interest Statement

OS Shirihai is a co-founder and SAB member of Enspire Bio LLC, Senergy-Bio and Capacity-Bio, and when this study was conducted, he was serving as a consultant to LUCA-Science, IMEL, Epirium, Johnson & Johnson, Pfizer, and Stealth Biotherapeutics. M Liesa is a co-founder of Enspire Bio LLC and consultant to Capacity Bio.

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
