## [Reviewer comments · Life Science Alliance]

Life Science Alliance

Mitochondria isolated from lipid droplets (LD) in WAT reveal functional differences based on LD size

Alexandra Brownstein, Michaela Veliova, Rebeca Acín-Pérez, Frankie Villalobos, Anton Petcherski, Alberto Tombolato, Marc Liesa, and Orian Shirihai

DOI: <https://doi.org/10.26508/lsa.202301934>

Corresponding author(s): Orian Shirihai, University of California, Los Angeles

Review Timeline:

Submission Date:	2023-01-19
Editorial Decision:	2023-03-02
Revision Received:	2023-09-08
Editorial Decision:	2023-10-05
Revision Received:	2023-10-20
Accepted:	2023-10-23

Transaction Report:

March 2, 2023

Re: Life Science Alliance manuscript #LSA-2023-01934

Prof. Orian S. Shirihai
University of California, Los Angeles
Department of Molecular and Medical Pharmacology, David Geffen School of Medicine
650 Charles East Young Drive South
CHS 27-200
Los Angeles, CA 90095

Dear Dr. Shirihai,

Thank you for submitting your manuscript entitled "Mitochondria isolated from lipid droplets reveal functional differences based on lipid droplet size" to Life Science Alliance. The manuscript was assessed by expert reviewers, whose comments are appended to this letter. We invite you to submit a revised manuscript addressing the Reviewer comments.

Thank you for this interesting contribution to Life Science Alliance. We are looking forward to receiving your revised manuscript.

Sincerely,

B. MANUSCRIPT ORGANIZATION AND FORMATTING:

Reviewer #1 (Comments to the Authors (Required)):

Shirhai and coworkers report on a modified protocol to isolate peridroplet mitochondria (PDM) from white adipose tissue (WAT), which also improves the efficiency of PDM isolation from brown adipose tissue (BAT). They included Proteinase K treatment in the high speed centrifugation protocol to improve detachment of mitochondria from lipid droplets (LD), based on previous studies indicating that protein tethers are mostly responsible for attaching mitochondria to lipid droplets. By using this approach, the authors were able to demonstrate that BAT and WAT PDM differ in their attachment to lipid droplets and in their bioenergetic capacity. They found that WAT PDM have a lower respiratory and ATP synthesis capacity than WAT cytosolic mitochondria. In addition, a negative correlation between lipid droplet size and WAT PDM respiratory capacity was observed. The optimized protocol allows the isolation and bioenergetic characterization of different PDM populations and thus may help to better understand PDM diversity. However, some aspects remain vague and should be addressed in a revised version.

Major points:

1.
The authors do not provide convincing evidence that their Proteinase K maintains outer mitochondrial membrane integrity. They should check whether there is degradation and/or leakage of mobile intermembrane space proteins rather than only checking inner membrane-bound respiratory complex components.
2.
By quantifying the amount of proteins, the authors suggest that Proteinase K treatment increases the yield of WAT CM and PDM and BAT PDM. What is the effect on mitochondrial purity?
3.
Fig. 1H/I: It is not clear why the authors check the impact of Proteinase K +/- PMSF on mitochondria after freeze thaw cycles, which disrupt mitochondrial membranes.
4.
The authors concluded that the strength of PDM attachment to lipid droplets differ between WAT and BAT. Has this anything to do with differences in the size of lipid droplets or with the association of different lipid droplets/PDM populations with the ER?
5.
The authors used the XF96 flux analyzer for quantifying the OCR. However, the description of the experimental details is vague regarding substrates and normalization.
6.
Apart from differences in OCR, are there differences in fatty acid oxidation and/or esterification between WAT and BAT PDM? Does the size of lipid droplets matter in this context?
7.
Do the differences in LD size observed after isolation from BAT and WAT correlate to the LD size profile in adipocytes?
8.
The authors state that comparisons between multiple groups were done by one-way ANOVA and respective post-hoc tests. However, the paragraph does not reference the analysis used to check for Gaussian distribution of the data, a prerequisite for the analysis of all the parametric test performed in this paper (t-tests, one-way ANOVA and two-way ANOVA). In addition, the authors continuously combine data from technical and biological replicates throughout the whole manuscript.

Reviewer #2 (Comments to the Authors (Required)):

1. Summary

The manuscript by Brownstein et al. titled "Mitochondria isolated from lipid droplets reveal functional differences based on lipid droplet size" describes a novel isolation method of mitochondria attached to lipid droplets in white adipocytes. This is an extremely important finding for the field and presents interesting and well controlled data indicating functional differences of peri-lipid droplet mitochondria (PDM) in white and brown adipocytes. Moreover, the authors show that ATP synthesis and respiratory activity of PDM are inversely correlated with lipid droplet size isolated from WAT.

2. Main findings and their support by data

Next to the technical advance a key conclusion is that PDMs in WAT have reduced respiratory capacity compared to PDM from BAT. The authors further present evidence that the levels of respiratory complexes change in PDM depending on LD size specifically in WAT. Further there is a clear difference in respiratory complex abundance between WAT and BAT.

The data are of exceptional quality and the paper is written extremely well. All experiments are well controlled, tested for significance and are robust. This reviewer was particularly impressed by the careful controls for possible artifacts that might be caused through addition of protease and protease inhibitors.

3. Additional points

There are two minor points that should be addressed:

a. In Figure 1 the authors give prominent space to the isolated LDs that are stripped with the classical protocol, which works well in BAT but not in WAT. Since the protease stripping protocol is a central point of the paper it would be good to include an image of the condition PDM + ProtK from WAT. Since this has been quantified in Fig. 1J the data should be available.

b. The first paragraph of the manuscript is slightly confusing and could potentially improve by simplifying it. PDMs can be isolated from BAT by simple differential centrifugation and salt washes (Benador 2018). These protocols have so far not led to successful isolation of PDMs from WAT, which seem to have higher LD affinity than in BAT (the historical papers that are referred to should then be cited). This is, however, only a suggestion and no requirement for publication.

Reviewer #3 (Comments to the Authors (Required)):

ReviewLSA

The present paper describes a new method to isolate mitochondria associated with lipid droplets (Peridroplet mitochondria or PDM) vs cytosolic mitochondria (CM) from murine white adipose tissue. The paper describes a protease based extension of previous work in BAT of the same group of authors where PDM and CM mitochondria were isolated based upon mechanical separation (differential centrifugation). By treating the PDM containing LD fractions with a protease, the yield of PDM from WAT increased significantly (as did the yield in PDM from BAT). Using this novel method, characteristics of PDM and CM mitochondria were assessed in pools isolated from WAT and BAT, and for PDM, LD size dependent effects on mitochondrial bioenergetics were observed and quantified and put into perspective of previous observations in BAT. From their combined biochemical, bioenergetics and microscopy studies the authors conclude that WAT and BAT derived PDM differ in their strength to bind to LDs, and reciprocally differ in their bioenergetics profile, with lower respiratory capacity in WAT PDM than in WAT CM (while the opposite was found for BAT) and higher ATP synthesis capacity in WAT derived PDM tethers to small LDs and lower ATP synthesis capacity in WAT-derived PDMs associated to larger LD's.

This paper elegantly introduces new methodology which permits evaluation of mitochondrial bioenergetics in mito's isolated from different pools and linked to differentially sized LDs. The rationale is clear, most of the methodology applied appears to be correct and valid (or its validity has at least been tested) and data appear to be collected and represented carefully. Data interpretation sometimes leaves room for alternatives that have not (yet) been discussed. I will try to outline some of my concerns/thoughts on the data interpretation below.

Overall, I am of the opinion that with some (relatively minor) revisions, this field will benefit from publication of this paper.

Major comments

1. The addition of ProtK and the higher yield of PDM mitochondria from WAT after using ProtK indeed indicates that in WAT mito's bind stronger to LDs than in BAT, but also the yield in BAT after using ProtK is higher than after relying on centrifugation solely. So apparently the protease is disrupting a protein-protein interaction that is present in both WAT and BAT. In Line 110 you are suggesting that your observations highlight the possibility that the mechanism of interaction between PDM and LD's might differ in WAT from BAT. But based upon the observation that ProtK also increases yield in BAT relative to centrifugation only, couldn't one also conclude that the type of protein-protein interaction might be the same, but the strength is different possibly simply by have more of the proteins that interact in WAT than in BAT? Please discuss.

2. To evaluate the 'amount of mitochondria per LD' (line 180), you compute the ration of mitotracker Deep Red derived signal over Bodipy 493 derived signal in a plate reader. If I recall correctly the intensity of the MTDR signal is a reflection of mitochondrial membrane potential whereas the bodipy signal just indicates that the lipid soluble dye is indeed dissolved in the LD, without changes in intensity. So the MTDR signal is not simply reflecting the 'amount' of mitos, but also the activity. Please

take this into account in the interpretation of the data. It would be of added value if the data are being confirmed using another, non-membrane potential dependent, mitochondrial dye.

3. While the use of the ProtK inhibitor PMSFs seems to suggest that the most of the mitochondrial isolated via the new protocol still seem intact and viable, the classical test to examine membrane leakage for isolated mitochondria (adding Cytochrome C to the system and hope to not see an increase in respiration) was not applied. Please explain why (possibly there is no added value of using Cyt C in a seahorse based system, in that case just explain in your rebuttal to this reviewer).

4. The hypothesized size dependency of PDM bioenergetics was tested and observed but solely for Complex I related substrates (pyruvate and malate) and not for palmitoylcarnitine or other Complex II dependent substrates. Do you happen to have an explanation for this and/or did you also test other fatty acids or succinate? Please discuss the substrate dependent size dependency.

5. On line 423 the authors state that 'in WAT PDM attached to smaller LD have a higher respiratory capacity' although not explicitly stated, this way of framing suggest that it is the higher respiratory capacity that makes the PDM bind preferentially to smaller LD's, while one can also argue that by having a higher respiratory capacity, the substrate fueling the oxidation are fatty acids originating from the LD and hence render the LD smaller. The fact that the size dependency of respiratory capacity was only detected on pyruvate and malate, on the other hand, might plead against this. Please discuss in more detail and if possible, provide data or references underpinning what is chicken and egg here (smaller LD's due to tethering with mito's with high respiratory capacity vs. preferentially binding of mitos with a high resp capacity to small LDs).

6. Line 426: I have never understood the reasoning in the original Benador paper that PDMs in BAT have a high capacity to generate ATP to facilitate the larger capacity for de novo lipogenesis in BAT and are therefore found attached to larger LD's. First and for all, the primary role of mito's in BAT is not to generate ATP, but rather to generate heat. Second, I might be incorrect, but I have never seen convincing evidence that de novo lipogenesis plays a major role in BAT TAG synthesis. I have always been under the impression that the majority of TAG in BAT originates from the (less ATP demanding) MAG, DAG re-esterification. Obviously, this is not the place to discuss/defend previously published papers, but it could help the reader of the current paper to provide a bit more underpinning of this statement. Also, if the BAT derived PDM indeed primarily serve to generate ATP, I would anticipate these mitochondria to have lower UCP1 levels than CM in BAT. Have you been in the position to explore this?

Minor comments

Title: I think the mitochondria are NOT isolated FROM the lipid droplets (which to me suggests that the mito's before isolation were IN the droplet, which they are not). Rather, I think you isolated mitochondria BOUND TO lipid droplets or cytosolic pools and that these pools have different bioenergetic profiles depending on the size of the LD (and on being derived from WAT vs BAT).

Reviewer #1 (Comments to the Authors (Required)):

Major points:1.

The authors do not provide convincing evidence that their Proteinase K maintains outer mitochondrial membrane integrity. They should check whether there is degradation and/or leakage of mobile intermembrane space proteins rather than only checking inner membrane-bound respiratory complex components.

Response: We thank the reviewer for the valuable comment. We have followed the reviewers suggestion and conducted the recommended experiment to check if there is leakage of intermembrane space components and degradation of mobile intermembrane space proteins. One of the classical tests for examining the integrity of outer membrane, is to add the electron carrier cytochrome c, a protein that lives in the intermembrane space, to the system and observe if there is an increase in respiration. We have included this experiment for WAT CM and PDM below, and added the results from all of the groups in supplementary figure 2. In the figure below, we see a sequential increase in state 3 respiration with the addition of both 10 µg/mL of cytochrome c and 100 µg/mL of cytochrome c. However, the fold change in state 3 respiration observed between the cytochrome c treatments is similar between the untreated PDM and the proteinase K treated PDM. This suggests the mitochondrial membrane integrity is similarly intact even after proteinase K treatment. The reason we show the fold increase is because cytochrome c can produce an artificial increase in OCR that does not come from mitochondria, and we show this artifact is equal between the control and the proteinase K treated mitochondria.

Furthermore, we see a similar sequential increase in oxygen consumption by cytochrome c addition even in the presence of Azide indicating there are non-mitochondrial mechanisms that are being affected by the treatment with cytochrome c, and not by cytochrome c directly replenishing the electron transport chain that is resulting in an increase in oxygen consumption.

Revised text (page 11):

“Because we observed that Prot K treatment selectively increased ATP synthesizing respiration in some mitochondrial preparations, it was still possible that Prot K induced a mild damage to the outer membrane, to cause mild cytochrome c leakage. If mild leakage was present, we should see a larger increase in respiration induced by cytochrome c supplementation in mitochondria isolated with proteinase K, when compared to cytochrome c

supplementation in mitochondria isolated without proteinase K. It is important to note that cytochrome c supplementation can increase oxygen consumption independently of complex IV activity as well. We found that cytochrome c supplementation did not induce a larger increase in respiration in Prot K treated mitochondria, with this effect being even of lower magnitude than the effect in control mitochondria (Supplementary Figure 2A). In this regard, the largest portion of cytochrome c-induced increase in respiration observed in both control and Prot K treated mitochondria was preserved in azide-treated mitochondria (Supplementary Figure 2B). This latter result shows that cytochrome c-induced increase in respiration is not caused by an increase in the availability of cytochrome c for mitochondrial respiration. These new data show that proteinase K treatment did not induce a mild damage in the outer membrane to cause a small leak in cytochrome c.”

2. By quantifying the amount of proteins, the authors suggest that Proteinase K treatment increases the yield of WAT CM and PDM and BAT PDM. What is the effect on mitochondrial purity?

Response: We thank the reviewer for this important question, which is addressed with the following data. In the revised manuscript we included Western blot data to show that an equal amount of several mitochondrial proteins, namely SDHA, ATP5A, UQCRC2, MTCO1 and VDAC are present in the proteinase K-isolated mitochondrial fraction per microgram of protein loaded. As all these mitochondrial proteins showed the same behavior, we can conclude that the purity obtained in isolated mitochondrial fractions using proteinase K is similar to classical protocols. We show representative Western blot assays in Figure 1H and 1I and their quantifications in Supplementary Figure 1.

[Figure removed by editorial staff per authors' request]

3.Fig. 1H/I: It is not clear why the authors check the impact of Proteinase K +/- PMSF on mitochondria after freeze thaw cycles, which disrupt mitochondrial membranes.

Response: We thank the reviewer for bringing up this question. The need of long-term storage of isolated mitochondria was the motivation justifying the addition of PMSF after mitochondrial isolation. Long term storage enables additional biochemical analyses of mitochondrial fractions that could not be done on the same day of isolation. As the reviewer pointed-out, freeze-thawing creates holes in the membranes and gives proteinase K access to all mitochondrial proteins for degradation. Thus, we aimed to establish conditions that inactivated proteinase K after isolation, to prevent the degradation of all mitochondrial proteins after a freeze-thaw cycle. We tested whether the addition of PMSF could prevent protein degradation of mitochondria isolated with proteinase K after a freeze-thaw cycle. Our data showed that PMSF was sufficient to preserve the same levels of mitochondrial proteins observed in mitochondrial fractions isolated without proteinase K. Therefore, by adding PMSF, we established an experimental approach that enables the long-term storage of our mitochondria at -80 °C for future analysis of mitochondrial matrix and inner membrane proteins.

4. The authors concluded that the strength of PDM attachment to lipid droplets differ between WAT and BAT. Has this anything to do with differences in the size of lipid droplets or with the association of different lipid droplets/PDM populations with the ER?

Response: We thank the reviewer for bringing up this important point. In the revised version we have included new text that answers this question. As a summary, we discuss that data presented in this manuscript supports that lipid droplet size can be a major determinant of the strength of PDM attachment. To emphasize this point, we have added these sentences in the discussion.

Revised text (page 23):

“When PDM from large and small LDs are isolated together, as they were in our initial experiments, the functional differences between the PDM from different populations are averaged together, concealing their unique properties. The new isolation protocol presented here allows us to explore open questions related to mitochondria-LD interaction and will help us better understand PDM function within individual cells and between different adipose tissue depots. Overall, we show data supporting the existence of at least two different populations of PDM, distinguished by their association with LDs of varying size. The differences in mitochondrial function observed between the subpopulations potentially reflect local LD environments with specific needs. We show that in order to isolate PDM from WAT, Proteinase K treatment is needed regardless of the size of the LD we are isolating PDM from. As the yield of PDM isolated from SM-LD and LG-LD by centrifugation is similar, we can conclude that the strength of PDM attachment to different size LDs is similar in BAT. We do however, observe more PDM attached to smaller LDs in WAT, and the opposite in BAT. This discrepancy between BAT

and WAT supports that other tissue-specific factors beyond lipid droplet size determine mitochondria-LD interaction.”

5. The authors used the XF96 flux analyzer for quantifying the OCR. However, the description of the experimental details is vague regarding substrates and normalization

Response: We thank the reviewer for pointing the need to clarify the experimental details. We have now added a new paragraph that expands the methods section and clarifies the protocol used to measure OCR. We also include the substrates and normalization used in the different assays in the figure legends, as well as in the methods sections. The methods section now reads:

Isolated Mitochondria Respirometry

“To calculate the effects of Prot K on oxygen consumption rates, we measured state 3 or the capacity of coupled mitochondria to produce ATP in the presence of substrates and ADP, oligomycin resistant leak and ATP-linked respiration, Complex IV-driven respiration and non-mitochondrial oxygen consumption.

Fold changes in respiration were calculated from raw OCR values obtained in each experimental group, where respiration of treated mitochondria isolated in the same day and analyzed on the same plate were normalized to the untreated mitochondria to establish the effect of PK and PMSF treatment. The individual fold changes over untreated mitochondria obtained from independent experiments were then averaged, represented in bar graphs. For each substrate (pyruvate malate or palmitoyl carnitine) and mitochondrial state (ie State 3, leak, ATP-linked, CoxIV) we calculated the fold change of each individual mitochondrial state between the mitochondria treated with PK and/or PMSF and

compared it to the untreated CM or PDM of the group. For example for State 3, we set the CM untreated equal to 1 by dividing by the raw OCR value and then compared the raw OCR values of the 3 other groups to the CM untreated to calculate the fold change. For the comparison between CM Prot K and PDM Prot K as in Figure 2G and Figure 3G, the fold changes were calculated from the raw OCR values of each individual experiment by comparing the OCR values of each state with the CM untreated state 3, which was set to 1.”

6. Apart from differences in OCR, are there differences in fatty acid oxidation and/or esterification between WAT and BAT PDM? Does the size of lipid droplets matter in this context?

Response: We thank the reviewer for pointing this out. We followed the reviewer's suggestion and have included measurements of palmitoyl-carnitine fueled oxygen consumption of PDM isolated from large and small lipid droplets, thus providing fatty acid oxidation capacity measurements in Figures 5B and D. We show that fat oxidation capacity is higher in PDM isolated from large lipid droplets in BAT, when compared to PDM from smaller BAT lipid droplets. On the other hand, we did not see differences in fat oxidation capacity of WAT PDM from large and small lipid droplets. Given that large lipid droplet fraction in BAT contain lipid droplets that are smaller than the large lipid droplet of the WAT, we conclude that fat oxidation capacity is maximal at a certain lipid droplet size and does not further increase after a certain threshold of size. Moreover, it has been shown that WAT do not oxidize fat in vivo, and have lower expression of fat-oxidizing enzymes. We have not developed an assay to measure esterification in fat layers isolated from tissue, as this is a method that might require an additional study.

[Figure removed by editorial staff per authors' request]

7. Do the differences in LD size observed after isolation from BAT and WAT correlate to the LD size profile in adipocytes?

Response: We thank the reviewer for the question. The lipid droplet sizes we observed in isolated fat layer from mouse WAT and BAT are consistent with what has been reported in vivo in previous studies. In isolated fat layers, we obtain lipid droplets from WAT with a perimeter varying between 1-230 μm , while lipid droplets are between 2-24 μm in BAT. In the literature, most report lipid droplet size as diameter, which is calculated by π (3.14) times the perimeter. If we compare our data with the literature, this is still consistent. The average lipid droplet diameter in WAT has been reported to be between 50-100 μm , which is approximately no larger than around 300 μm which we have seen in our data, but because our values are averaged we are not showing the maximum in our graphs. In BAT has been reported to be <10 μm , which is approximately <31.4 and falls within the range we have observed and are reporting (PMID: 28793320, PMID: 30652361).

[Figure removed by editorial staff per authors' request]

8. The authors state that comparisons between multiple groups were done by one-way ANOVA and respective post-hoc tests. However, the paragraph does not reference the analysis used to check for Gaussian distribution of the data, a prerequisite for the analysis of all the parametric test performed in this paper (t-tests, one-way ANOVA and two-way ANOVA). In addition, the authors continuously combine data from technical and biological replicates throughout the whole manuscript.

Response: We thank the reviewer for pointing out that we did not clearly indicate the normality tests used. We have now updated the methods section that outlines the normality tests used, as well as the corresponding parametric and non-parametric tests used in each set of analyses. Moreover, we clarified that we only combined the data of independent mitochondrial isolations, which makes the presentation of the data and the statistical analyses less heterogeneous.

Statistical analysis

“Statistical analyses were performed using GraphPad Prism 5.03 (GraphPad Software Inc., San Diego, CA, USA). Data were presented as mean \pm SEM for all conditions. Normality of data was checked by Kolmogorov-Smirnov test with the Dallal-Wilkinson-Lillie for corrected p value. For data with normal distribution, one-way ANOVA with Šídák multiple comparisons test was employed. Individual means were compared using the parametric two-tail Student’s t-test. For non-parametric data, Kruskal-Wallis with Dunn’s multiple comparisons test was used. Comparisons between two groups were assessed by Kolmogorov-Smirnov test, and when appropriate Wilcoxon matched-pairs rank test. Differences of $P < 0.05$ were considered to be significant. All graphs and statistical analyses were performed using GraphPad Prism 9 (GraphPad Software, San Diego, CA).”

Reviewer #2 (Comments to the Authors (Required)):

There are two minor points that should be addressed:

a. In Figure 1 the authors give prominent space to the isolated LDs that are stripped with the classical protocol, which works well in BAT but not in WAT. Since the protease stripping protocol is a central point of the paper it would be good to include an image of the condition PDM + ProtK from WAT. Since this has been quantified in Fig. 1J the data should be available.

Response: We thank the reviewer for identifying this oversight. We have updated Figure 1A to include images of isolated LDs treated with Prot K.

[Figure removed by editorial staff per authors' request]

b. The first paragraph of the manuscript is slightly confusing and could potentially improve by simplifying it. PDMs can be isolated from BAT by simple differential centrifugation and salt washes (Benador 2018). These protocols have so far not led to successful isolation of PDMs from WAT, which seem to have higher LD affinity than in BAT (the historical papers that are referred to should then be cited). This is, however, only a suggestion and no requirement for publication.

Response: We thank the reviewer for the suggested changes to improve the clarity of the first paragraph. We have modified the first paragraph of the text to be less confusing. Now it reads as:

“Recent studies in brown adipose tissue (BAT) described a unique subpopulation of mitochondria bound to lipid droplets (LDs), peridroplet mitochondria (PDM). PDMs can be isolated from BAT by differential centrifugation and salt washes (Benador *et al*, 2018). Contrary to BAT, this approach has so far not led to successful isolation of PDMs from WAT. Here, we developed a method to isolate PDM from WAT with high yield and purity by an optimized proteolytic treatment that preserves the respiratory function of mitochondria. Using this approach, we show that, contrary to BAT, WAT PDM have lower respiratory and ATP synthesis capacity compared to WAT CM. Furthermore, by isolating PDM from LDs of different sizes, we find a negative correlation between LD size and the respiratory capacity of their PDM in WAT. Thus, our new isolation method reveals tissue-specific characteristics of PDM and establishes the existence of heterogeneity in PDM function determined by LD size.”

Reviewer #3 (Comments to the Authors (Required)):

ReviewLSA

The present paper describes a new method to isolate mitochondria associated with lipid droplets (Peridroplet mitochondria or PDM) vs cytosolic mitochondria (CM) from murine white adipose tissue. The paper describes a protease based extension of previous work in BAT of the same group of authors where PDM and CM mitochondria were isolated based upon mechanical separation (differential centrifugation). By treating the PDM containing LD fractions with a protease, the yield of PDM from WAT increased significantly (as did the yield in PDM from BAT). Using this novel method, characteristics of PDM and CM mitochondria were assessed in pools isolated from WAT and BAT, and for PDM, LD size dependent effects on mitochondrial bioenergetics were observed and quantified and put into perspective of previous observations in BAT. From their combined biochemical, bioenergetics and microscopy studies the authors conclude that WAT and BAT derived PDM differ in their strength to bind to LDs, and reciprocally differ in their bioenergetics profile, with lower respiratory capacity in WAT PDM than in WAT CM (while the opposite was found for BAT) and higher ATP synthesis capacity in WAT derived PDM tethers to small LDs and lower ATP synthesis capacity in WAT-derived PDMs associated to larger LD's.

This paper elegantly introduces new methodology which permits evaluation of mitochondrial bioenergetics in mito's isolated from different pools and linked to differentially sized LDs. The rationale is clear, most of the methodology applied appears to be correct and valid (or its validity has at least been tested) and data appear to be collected and represented carefully. Data interpretation sometimes leaves room for alternatives that have not (yet) been discussed. I will try to outline some of my concerns/thoughts on the data interpretation below. Overall, I am of the opinion that with some (relatively minor) revisions, this field will benefit from publication of this paper.

Major comments

1. The addition of ProtK and the higher yield of PDM mitochondria from WAT after using ProtK indeed indicates that in WAT mito's bind stronger to LDs than in BAT, but also the yield in BAT after using ProtK is higher than after relying on centrifugation solely. So apparently the protease is disrupting a protein-protein interaction that is present in both WAT and BAT. In Line 110 you are suggesting that your observations highlight the possibility that the mechanism of interaction between PDM and LD's might differ in WAT from BAT. But based upon the observation that ProtK also increases yield in BAT relative to centrifugation only, couldn't one also conclude that the type of protein-protein interaction might be the same, but the strength is different possibly simply by have more of the proteins that interact in WAT than in BAT? Please discuss.

Response: We thank the reviewer for this suggestion. We have now acknowledged this possibility raised by the reviewer by including new text in the discussion.

Revised text (page 18):

“Moreover the difference in strength between mitochondria and LDs can be mediated either by a different composition of protein tethers, as supported by differences in PLIN5 and MIGA2 between BAT and WAT, and/or by a larger number of other uncharacterized tethers shared between WAT and BAT (Freyre *et al.*, 2019).”

2. To evaluate the 'amount of mitochondria per LD' (line 180), you compute the ration of mitotracker Deep Red derived signal over Bodipy 493 derived signal in a plate reader. If I recall correctly the intensity of the MTDR signal is a reflection of mitochondrial membrane potential whereas the bodipy signal just indicates that the lipid soluble dye is indeed dissolved in the LD, without changes in intensity. So the MTDR signal is not simply reflecting the 'amount' of mitos, but also the activity. Please take this into account in the interpretation of the data. It would be of added value if the data are being confirmed using another, non-membrane potential dependent, mitochondrial dye.

Response: The thank the reviewer for bringing out this very important point. When we first described respirometry in previously frozen samples (PMID: 32432379), we also proposed a method to express the respiratory rates by mitochondrial content. We tried different dyes such as mitotracker red (MTR), TMRE, and mitotracker deep red (MTDR). While TMRE and MTR were membrane potential dependent, MTDR was not. This different behavior allowed us to use MTDR in previously frozen samples as well as fixed samples where there is no membrane potential involved. We are including below some of the panels used in our previous publication to illustrate these results, referring to the original Figure panels used in PMID: 32432379. We have also clarified this in the text.

Revised text (page 9):

“We chose to use MTDR because we have previously verified its membrane potential dependency is minimal and does not impose a significant bias over mitochondrial mass with the concentration and duration of staining in our protocol (Acín-Perez *et al.*, 2021).”

3. While the use of the ProtK inhibitor PMSFs seems to suggest that the most of the mitochondrial isolated via the new protocol still seem intact and viable, the classical test to examine membrane leakage for isolated mitochondria (adding Cytochrome C to the system and hope to not see an increase in respiration) was not applied. Please explain why (possibly there is no added value of using Cyt C in a seahorse based system, in that case just explain in your rebuttal to this reviewer).

Response: We thank the reviewer for this suggestion, which was also raised by reviewer 1. We performed new experiments in which the effects of cytochrome c supplementation on respiration of PDM isolated using the classical protocol were compared to PDM isolated using proteinase K. We found that the effects on respiration induced by cytochrome c supplementation were similar between mitochondria isolated with and without proteinase K, which rules out the possibility that proteinase K supplementation induces cytochrome c depletion. Therefore, this result demonstrates that proteinase K does not induce outer membrane leakage and subsequent cytochrome c loss.

Revised text (page 11-12):

“Because we observed that Prot K treatment selectively increased ATP synthesizing respiration in some mitochondrial preparations, it was still possible that Prot K induced a mild damage to the outer membrane, to cause mild cytochrome c leakage. If mild leakage was present, we should see a larger

increase in respiration induced by cytochrome c supplementation in mitochondria isolated with proteinase K, when compared to cytochrome c supplementation in mitochondria isolated without proteinase K. It is important to note that cytochrome c supplementation can increase oxygen consumption independently of complex IV activity as well. We found that cytochrome c supplementation did not induce a larger increase in respiration in Prot K treated mitochondria, with this effect being even of lower magnitude than the effect in control mitochondria (Supplementary Figure 2A). In this regard, the largest portion of cytochrome c-induced increase in respiration observed in both control and Prot K treated mitochondria was preserved in azide-treated mitochondria (Supplementary Figure 2B). This latter result shows that cytochrome c-induced increase in respiration is not caused by an increase in the availability of cytochrome c for mitochondrial respiration. These new data show that proteinase K treatment did not induce a mild damage in the outer membrane to cause a small leak in cytochrome c.”

4. The hypothesized size dependency of PDM bioenergetics was tested and observed but solely for Complex I related substrates (pyruvate and malate) and not for palmitoylcarnitine or other Complex II dependent substrates. Do you happen to have an explanation for this and/or did you also test other fatty acids or succinate? Please discuss the substrate dependent size dependency.

Response: This is a very important point and we thank the reviewer for his/her comments. The yield of PDM mitochondria isolated from WAT is significantly smaller than that of BAT, therefore we had to prioritize the parameters tested. We previously published the comparison between the oxygen consumption rates of BAT peridroplet mitochondria (PDM) and cytosolic mitochondria (CM) under succinate-rotenone, pyruvate and malate and palmitoyl carnitine (PMID: 29617645). In Benador 2018, PDM had increased state III and maximal respiration compared to CM on either pyruvate

malate and succinate rotenone. Because of sample limitations we prioritized comparing pyruvate malate respiration and palmitoyl carnitine, the two fuels that showed differential utilization between PDM and CM in BAT.

Consequently, we prioritized palmitoyl-carnitine over other fatty acids, as palmitoyl-carnitine was the substrate originally used to characterize PDM in the first publication, compared side to side with pyruvate and malate (PMID: 29617645). As a major goal of this paper is to determine whether isolation with Prot K preserved PDM functionality as published, we believe that trying other fatty acids different than palmitoyl carnitine is beyond the scope of this paper. This leaves some potential differences to be tested in future studies.

[Figure removed by editorial staff per authors' request]

Quantification of respiratory states driven by succinate/rotenone in cytoplasmic (blue) and peridroplet (red) mitochondria. State III quantifies respiration driven by ATP synthesis and maximal respiration quantifies maximal electron transport activity induced by the chemical uncoupler FCCP. 6 technical replicates per group. N = 7 independent isolations. For each individual experiment, average OCR values of CM and PDM were normalized to the average OCR of all mitochondria (see Quantification and Statistical Analysis for complete equations). Data are expressed as means \pm SEM. * $p < 0.0001$.**

5. On line 423 the authors state that in WAT PDM attached to smaller LD have a higher respiratory capacity' although not explicitly stated, this way of framing suggest that it is the higher respiratory capacity that makes the PDM bind preferentially to smaller LD's, while one can also argue that by having a higher respiratory capacity, the substrate fueling the oxidation are fatty acids originating from the LD and hence render the LD smaller. The fact that the size dependency of respiratory capacity was only detected on pyruvate and malate, on the other hand, might plead against this. Please discuss in

more detail and if possible, provide data or references underpinning what is chicken and egg here (smaller LD's due to tethering with mito's with high respiratory capacity vs. preferentially binding of mitos with a high resp capacity to small LDs).

Response: We thank the reviewer for this very insightful comment and apologize for not including a discussion on this. From the current experiments we cannot say whether the higher respiratory capacity induces PDM to bind preferentially to smaller LDs or if the higher respiratory capacity of PDMs fueling the oxidation of fatty acids leads to the observation of more active PDM on smaller LDs in WAT.

With the exception of beige adipocytes, WAT depots are not fat oxidizing tissues. This suggests that when mitochondria interact with small LDs in WAT, they are not oxidizing these fatty acids. At the same time, it is known that small LDs are the ones that can expand more and are thus more active esterifying and breaking lipids. While our experiments do not exclude this possibility, we believe that in this case, the interaction of PDM with small LDs, and the higher capacity for ATP-synthesis supports a higher activity of TG esterification in small LDs compared to large LD.

Revised text (page 21-22):

“We show that fat oxidation capacity is higher in PDM isolated from large lipid droplets in BAT, when compared to smaller BAT lipid droplets. On the other hand, we did not see differences in fat oxidation capacity of WAT PDM from large and small lipid droplets. Remarkably, large-lipid droplet fraction in BAT contain lipid droplets of the same size as small lipid droplet fraction in WAT. This may reflect a relationship between fat utilization efficiency and LD size, where fat oxidation capacity is maximal at a certain lipid droplet size and does not further increase after a certain threshold of size.

From the current experiments, we cannot definitively say whether the higher respiratory capacity induces PDM to bind preferentially to smaller LDs or if the higher respiratory capacity of PDMs fueling the oxidation of fatty acids leads to the observation of more active PDM on smaller LDs in WAT. However, fat oxidation is minimal in WAT. This suggests that when mitochondria interact with

small LDs in WAT, they are not oxidizing these fatty acids. At the same time, as previously described, the small LDs in WAT are the ones that can expand and are thus more active esterifying and breaking lipids. Therefore, we suggest that in WAT, the interaction of PDM with small LDs and the higher capacity for ATP-synthesis supports a higher activity of TG esterification in small LDs compared to large LD.”

6. Line 426: I have never understood the reasoning in the original Benador paper that PDMs in BAT have a high capacity to generate ATP to facilitate the larger capacity for de novo lipogenesis in BAT and are therefore found attached to larger LD's. First and for all, the primary role of mito's in BAT is not to generate ATP, but rather to generate heat. Second, I might be incorrect, but I have never seen convincing evidence that de novo lipogenesis plays a major role in BAT TAG synthesis. I have always been under the impression that the majority of TAG in BAT originates from the (less ATP demanding) MAG, DAG re-esterification. Obviously, this is not the place to discuss/defend previously published papers, but it could help the reader of the current paper to provide a bit more underpinning of this statement. Also, if the BAT derived PDM indeed primarily serve to generate ATP, I would anticipate these mitochondria to have lower UCP1 levels than CM in BAT. Have you been in the position to explore this?

Response: We thank the reviewer for raising this point and the importance of referring in more detail to the previous paper by Benador et al (PMID: 29617645). Benador et al. showed that not all mitochondria in BAT produce ATP (Figure 2 H and I), but only PDM have increased ATP-synthesizing capacity. In addition, Benador et al. showed that indeed PDM are a minority in brown adipocytes when thermogenesis is activated. This means that when thermogenesis is activated, most mitochondria are not producing ATP. Second, because previous studies assessing mitochondrial function in BAT discarded the fat layer, only cytosolic mitochondria were analyzed and PDM were thus not included. Therefore, previous studies did not measure PDM function, and that is why these references could not detect any ATP-synthesizing respiration in BAT.

The data in Benador et al. showed that PDM support LD expansion by providing ATP for fatty acid esterification into triglycerides, and not de-novo lipogenesis, as shown in Figure 6 of PMID: 29617645.

Although in the Benador et al. paper we do not see changes in UCP1 protein expression, as observed in Figure 4C of PMID: 29617645, ATP is a known inhibitor of UCP1, and the higher ATP-synthesizing capacity of PDM might be serving to produce ATP that blocks UCP1 activity, as well as facilitating the esterification of triglycerides.

Moreover, we have also analyzed UCP1 protein content in PDM versus CM of isolated mitochondria treated with proteinase K and PMSF, and we reproduce the absence of differences in UCP1 content between PDM and CM.

[Figure removed by editorial staff per authors' request]

Figure 6. Mitochondria-Lipid Droplet Association Promotes Triacylglyceride Synthesis
(A) Representative thin-layer chromatography (TLC) of cellular lipids extracted from cultured brown adipocytes untransduced (control), transduced with the full-length Plin5 that contains mitochondrial recruiting sequence (Plin5), and transduced with truncated Plin5 that lacks the mitochondria recruitment sequence (Plin5 Δ 399–463). Cells were incubated with BODIPY C12 558/568 (C12) overnight to assess triacylglyceride (TAG) synthesis. The mobility of fatty acid species from loading origin is determined by relative polarity, with TAG migrating the highest.

(B) Quantification of TAG from n = 3 independent experiments. Data were normalized to control for each individual experiment.

(C and D) TLC of cultured brown adipocytes incubated with C12 with or without the fatty acid esterification inhibitor Triacsin C (red) (C). In the histogram, note the decrease in TAG and increase in free C12 induced by Triacsin C (D).

(E) Representative TLC of cultured brown adipocytes incubated with C12 with or without the mitochondrial ATP synthase inhibitor oligomycin.

(F) Quantification of TAG synthesis dependent on mitochondrial ATP synthase activity from n = 3 independent experiments. OXPHOS-dependent TAG synthesis was calculated as the difference in TAG between oligomycin-treated and untreated cells.

Data are expressed as means \pm SEM. *p < 0.05, **p < 0.001.

October 5, 2023

RE: Life Science Alliance Manuscript #LSA-2023-01934R

Prof. Orian S. Shirihai
University of California, Los Angeles
Department of Molecular and Medical Pharmacology, David Geffen School of Medicine
650 Charles East Young Drive South
CHS 27-200
Los Angeles, CA 90095

Dear Dr. Shirihai,

Thank you for submitting your revised manuscript entitled "Mitochondria isolated from lipid droplets (LD) in WAT reveal functional differences based on LD size". We would be happy to publish your paper in Life Science Alliance pending final revisions necessary to meet our formatting guidelines.

- please address Reviewer 2's remaining comments
- please add the Twitter handle of your host institute/organization as well as your own or/and one of the authors in our system
- titles in the system and the manuscript file must match -- please correct
- please make sure the author order in your manuscript and our system match
- please mark the corresponding authors in the manuscript file
- please add your main, supplementary figure, and table legends to the main manuscript text after the references section;
- please consult our manuscript preparation guidelines <https://www.life-science-alliance.org/manuscript-prep> and make sure your manuscript sections are in the correct order
- please be sure to add all authors in the author's contribution section
- please add your main, supplementary figure, and table legends to the main manuscript text after the references section
- please add callouts for Figures S1A-D and S2C-D to your main manuscript text;

A. FINAL FILES:

B. MANUSCRIPT ORGANIZATION AND FORMATTING:

Sincerely,

Reviewer #1 (Comments to the Authors (Required)):

In the revised version of the manuscript, the authors have adequately addressed all issues.

Reviewer #2 (Comments to the Authors (Required)):

The comments by this reviewer have been addressed. I have a few comments to the revised version of the manuscript:

I apologize for not having seen the graphical abstract before, but the illustration indicates preservation of the outer mitochondrial membrane proteins and their interactors on LDs after PK treatment. This reviewer believes that PK degrades the cytoplasmic portion of polypeptides on organelle surfaces, and their degradation is necessary for the efficient release of mitochondria from LDs. The drawing is perhaps misleading in that it suggests preservation of the tethers during PK treatment. This is most certainly not the case. Please correct the drawing.

This reviewer suggests to discuss the possibility of a lipid droplet mitochondrial tethering mechanism that involves protein- lipid interaction. Biochemically speaking, it is not clear that a protein-protein interaction prevents the successful stripping of the WAT PDMs. It may well be that the stripping fails because of a strong hydrophobic interaction between a mitochondrial surface factor and the LD. A protein-protein interaction should be broken by addition of salt in the isolation buffer as observed for BAT PDMs.

All in all, this is a careful study on LD-mitochondria interaction and should be published without further delay!

Reviewer #3 (Comments to the Authors (Required)):

The authors have carefully considered most of my comments and dealt with it appropriately. In my opinion the revised version of the MS is ready for acceptance, no further comments/questions. Well done!

Reviewer #2 (Comments to the Authors (Required)):

The comments by this reviewer have been addressed. I have a few comments to the revised version of the manuscript:

I apologize for not having seen the graphical abstract before, but the illustration indicates preservation of the outer mitochondrial membrane proteins and their interactors on LDs after PK treatment. This reviewer believes that PK degrades the cytoplasmic portion of polypeptides on organelle surfaces, and their degradation is necessary for the efficient release of mitochondria from LDs. The drawing is perhaps misleading in that it suggests preservation of the tethers during PK treatment. This is most certainly not the case. Please correct the drawing.

Response: We thank the reviewer for the valuable comment, and we have corrected the graphical abstract. It now reflects that the cytoplasmic portion of polypeptides are degraded by proteinase K.

2. This reviewer suggests to discuss the possibility of a lipid droplet mitochondrial tethering mechanism that involves protein- lipid interaction. Biochemically speaking, it is not clear that a protein-protein interaction prevents the successful stripping of the WAT PDMs. It may well be that the stripping fails because of a strong hydrophobic interaction between a mitochondrial surface factor and the LD. A protein-protein interaction should be broken by addition of salt in the isolation buffer as observed for BAT PDMs.

All in all, this is a careful study on LD-mitochondria interaction and should be published without further delay!

Response: We thank the reviewer for bringing up this important point. In the revised version we have included new text that includes the possibility of both protein-protein and protein-lipid interactions that contribute to PDM-LD attachment and the strong attachment of PDM in the WAT.

Revised text (page 5-6):

A WAT-specific protein-protein or protein-lipid interaction may explain the stronger attachment of PDM in WAT versus BAT.

Our results show that mechanical separation effectively removed the majority, but not all PDM from lipid droplets in the BAT (**Fig 1C and 1D**). However application of the

same mechanical protocol to WAT did not result in any significant removal of PDM from lipid droplets (**Fig 1A and 1B**), suggesting the possibility that the composition and or/abundancy of tethers between mitochondria and lipid droplets are different in BAT and WAT. We hypothesized that the resistance to centrifugation may come from either protein-protein or protein-lipid interactions. Independent of which of the above mechanisms contribute to mitochondria-LD tethering, a protein is expected to be involved and therefore should be sensitive to proteolytic activity. To test our hypothesis, we treated the fat layers of WAT with Proteinase K (Prot K), to digest the protein-mediated tethers anchoring mitochondria to LDs and potentially strip mitochondria that are resistant to stripping by centrifugation.

Revised text:

Discussion (line 404-408)

The novelty of our new method is the treatment of lipid droplet fractions, containing PDM, with a protease. This step was inspired by previous studies supporting that protein-protein interactions and protein-lipid interactions are key tethering mechanisms (Boutant *et al*, 2017; Freyre *et al*, 2019; Stone *et al*, 2009; Wang *et al*, 2011).

October 23, 2023

RE: Life Science Alliance Manuscript #LSA-2023-01934RR

Prof. Orian S. Shirihai
University of California, Los Angeles
Department of Molecular and Medical Pharmacology, David Geffen School of Medicine
650 Charles East Young Drive South
CHS 27-200
Los Angeles, CA 90095

Dear Dr. Shirihai,

Thank you for submitting your Methods entitled "Mitochondria isolated from lipid droplets (LD) in WAT reveal functional differences based on LD size". It is a pleasure to let you know that your manuscript is now accepted for publication in Life Science Alliance. Congratulations on this interesting work.

DISTRIBUTION OF MATERIALS:

Again, congratulations on a very nice paper. I hope you found the review process to be constructive and are pleased with how the manuscript was handled editorially. We look forward to future exciting submissions from your lab.

Sincerely,
